

# Ammonia emission measurements of an intensively grazed pasture

Karl Voglmeier[1,2], Markus Jocher[1], Christoph Häni[3], Christof Ammann[1]

[1]Climate and Agriculture Group, Agroscope, Zürich, 8046, Switzerland
[2]Department of Environmental Systems Science, ETH Zurich, Zürich, 8092, Switzerland
[3]School of Agricultural, Forest and Food Sciences HAFL, Bern University of Applied Sciences, Zollikofen, 3052, Switzerland

*Correspondence to*: Karl Voglmeier (karl.voglmeier@agroscope.admin.ch)

**Abstract.** The quantification of ammonia ($NH_3$) emissions in ambient air conditions is still a challenge and the corresponding emission factor for grazed pastures have therefore a large uncertainty. This study presents $NH_3$ emission measurements of two pasture systems in western Switzerland over the entire grazing season 2016. During the measurement campaign, each pasture
system was grazed by 12 dairy cows in an intensive rotational management. The cow herds on the two pastures differed in the energy to protein balance of the diet. $NH_3$ concentrations were measured upwind and downwind of a grazed sub plot with line integrating open path instruments that were able to retrieve small horizontal concentration differences ($< 0.2 \mu g\ NH_3\ m^{-3}$). The $NH_3$ emission fluxes were calculated by applying a backward Lagrangian Stochastic (bLS) dispersion model to the difference of paired concentration measurements and prevailed within a range of 0 to 2.5 µg $N-NH_3\ m^{-2}\ s^{-1}$. The fluxes increased steadily
during a grazing interval from previous non-significant values to reach maximum emissions at the end of the grazing interval. Afterwards they decreased exponentially to near zero values within 3-5 days. A standard emission curve was calculated for each of the two systems and adopted to each rotation in order to account for missing data values and to estimate inflow disturbances due to grazing on upwind paddocks. Dung position measurements and cow position monitoring were performed to account for the non-negligible inhomogeneity of cow excreta on the pasture. The average emission (± std. dev. of individual
rotation values) per grazing hour was calculated as $0.64 \pm 0.11$ g $N-NH_3\ cow^{-1}\ h^{-1}$ for the herd with the N balanced diet (system M) and $1.07 \pm 0.06$ g $N-NH_3\ cow^{-1}\ h^{-1}$ for the herd with the protein rich grass-only diet (system G). Surveys of feed intake, body weight and milk yield of the cow herds were used to estimate the nitrogen (N) excretion by an animal N budget model. Based on that, mean relative emission factors of $6.4 \pm 2.0$ % and $8.7 \pm 2.7$ % of the applied urine N were found for the systems M and G, respectively. The results can be used to validate the Swiss national emission inventory and demonstrate the positive
effect of a N-balanced diet on pasture $NH_3$ emission.

## 1 Introduction

Agricultural livestock production is the main source of air pollution by ammonia ($NH_3$) (Bouwman et al., 1997). The largest share of the emissions is usually assigned to the excretions in the barn with subsequent manure storage and spreading (Kupper et al., 2015). The high emissions are largely responsible for the formation of secondary aerosols in the atmosphere through



reactions with nitric and sulfuric acids (Nemitz et al., 2009). This can have a significant effect on human health and can also lead to eutrophication and acidification of the environment through deposition (Sutton et al., 2011).

Grazing is considered as one efficient mitigation option to reduce $NH_3$ volatilisation due to the direct infiltration of urine in the soil before urea is degraded to ammonium and ammonia. According to the Swiss inventory model Agrammon (Kupper et al., 2015) grazing of cattle produces about eight times lower emissions compared to indoor housing (including storage and spreading of manure). Emission inventories usually make use of generalized emission factors that relate emissions to the corresponding source of water soluble nitrogen (urea, ammonium or dissolved ammonia). In the case of grazed pastures the relevant nitrogen (N) source is the input of urine by animal excretion (Petersen et al., 1998). However the pasture emission factor still has a large uncertainty because corresponding $NH_3$ emission experiments are rare and the available studies reported a large range of emission factors (5 to 25.7 % of excreted urine N; see e.g. Jarvis et al., 1989, Bussink, 1992, Laubach et al., 2012, 2013b). Many of the studies used manual applied urine and measured the emissions with chamber or wind tunnel methods. These techniques might lead to questionable results due to the altering of the environment and the high heterogeneity of the emissions (Misselbrook et al., 2005; Sintermann et al., 2012).

Volten et al. (2012) introduced a new open path miniDOAS system that measures line integrated ammonia concentrations with a relatively high temporal resolution. Sintermann et al. (2016) adopted and developed the system to field applicability and suggested that paired miniDOAS systems in combination with a dispersion model can be used to estimate emissions of a pasture. Bell et al. (2017) estimated the $NH_3$ emission factor based on miniDOAS concentration measurements in combination with a backward Lagrangian Stochastic (bLS) dispersion model for a 12 day period and demonstrated the applicability of the miniDOAS / bLS combination for grazing systems. However no information on the excreta distribution on the pasture was obtained and retrieved emission factors were based on a standard cow and feeding strategy. The relatively short measurement campaign in May also limited the representativeness of the derived emission factor for a full year. For micrometeorological methods a spatially homogenous source area is usually needed (Munger et al., 2012) which is often not the case on grazed pastures (Draganova et al., 2016). However only very few studies reported on the uncertainty associated with a heterogeneous emission source and those studies usually focused on greenhouse gas emissions (Felber et al., 2015; Peltola et al., 2015).

In the present experiment the miniDOAS systems in combination with bLS modelling were applied to determine $NH_3$ emissions of two paired rotational grazing system over a full grazing season. Position monitoring of dung patches with GPS and of cows with a camera system were used to relate the measured emissions to the animal and excreta density. The calculated emission factors were based on actual in situ cow productivity data and feed analyses and were compared to standard emission factors.



## 2 Material and methods

### 2.1 Site description and experimental design

The study site was located in the Pre-Alps of Switzerland at the research farm Agroscope Posieux in the canton of Fribourg (46°46´04´´N, 7°06´28´´E). The soil is classified as stagnic Anthrosol with a loamy texture (20 % clay, 35 % silt and 45 %

sand) and the vegetation consisted mainly of a typical grass clover mixture (10 % to 50 % *Lolium perenne* and 7 % to 40 % *Trifolium repens*) with an increasing clover share during the grazing season. In 2007 the last renovation of the site took place. Since then the site has been used as an intensive pasture for cattle. Averaged over the past years, the average fertilizer application rate was about 120 kg N ha$^{-1}$ per year, in addition to the excreta of grazing animals. Climate records show an annual average temperature of 8.7 °C and an annual precipitation amount of 1075 mm (MeteoSwiss, 2018). The experiment was

conducted at a 5.5 ha pasture and the cows were managed in a rotational grazing system (see Fig. 1). The whole pasture was divided into two separate systems having different feeding strategies of the cows. The southern system (labeled "G") represented a full grazing regime without additional feed supplementation. This resulted in a considerable protein surplus for the animals leading to an unnecessary high N excretion. At the northern system (labeled "M") cows were provided with additional maize silage (roughly 25 % of the total feed dry matter intake) which has a low protein content and resulted in a

more demand-adjusted optimized protein content in the diet (see also Arriaga et al., 2010; Yan et al., 2006) leading to less N excretion. Each of the two pasture systems was divided into 11 paddocks resulting in a full rotation period of about 20 days, depending on the grass growth conditions. The size of the paddocks were adjusted to the different treatments: 1700 m$^2$ for the northern M system and 2200 m$^2$ for the southern G system.  The grazing rotation was synchronous for the two systems and started in the middle of the fields (on paddocks X.11 with X indicating both fields) in westerly direction (until paddock X.16)

and then from the middle (X.21) to the eastern side of the field (X.25). Twice a day (around 05:00 – 07:00 and 15:00 – 17:00 LT) the cows were brought to the nearby barn for milking. However, in cases of high air temperatures in August and beginning of September the cows spent a longer period in the barn during daytime (typically 11:00 – 17:00 LT). Due to dry periods during the summer month and subsequent low grass growth additional pasture areas were used for grazing. The herd for each system consisted of 12 dairy cows. The main measurement campaign took place between May and October 2016, and in

summary, seven full grazing rotations took place in that period (see Table 1). During the measurement campaign, the site was fertilized with ammonium nitrate (28 kg ha$^{-1}$, end of June) and urea (42 kg ha$^{-1}$, X.11–X.16 mid of August, X.21–X.25 beginning of September).

### 2.2 Ammonia emission measurements

#### 2.2.1 Ammonia concentration

Line-integrated ammonia concentrations were measured using four miniDOAS systems (Sintermann et al., 2016). These open path instruments make use of the differential optical absorption in the UV range (200 – 230 nm). Two miniDOAS systems (namely MD5 and MD2, naming based on serial number) were installed at system M and two instruments (MD1 and MD6)



on system G (Fig. 1a). All instruments were installed at a height of about 1.3 m. Each miniDOAS pair (e.g. MD5 and MD2) was separated by a horizontal distance of about 30 m which allowed for concentration measurements upwind and downwind of a subplot of the paddocks in between. The single light path between the sensor and the retroreflector for the individual devices had a length of 30 to 35 m. The instruments reported $NH_3$ concentration at a temporal resolution of one minute. The one minute data were averaged to 30 minute values for further processing. Due to the predominant wind directions NE and SW one miniDOAS usually reported upwind concentration $C_{Upwind}$ and the other one the downwind concentration $C_{Downwind}$ (see Fig. 1). This setting allowed for the computation of the horizontal concentration gradient $\Delta C$ caused by emissions from the area in between. The important reference spectrum (see Sintermann et al., 2016) for each miniDOAS was determined during a seven day inter-comparison campaign at the Chaumont, Switzerland (47°02′58′′N, 6°58′16′′E, 1136m, 20-27 July 2016). The site is located 30 km north-west of Posieux and is only marginally contaminated by $NH_3$ and was therefore ideal to compute the reference spectra. The miniDOAS systems were operated in parallel and compared to wet chemical impingers (see Häni et al., 2016) in order to retrieve the instrumental offset and absolute concentration.

### 2.2.2 Turbulence and meteorological parameters

For the characterization of turbulent mixing the three dimensional wind velocity (u,v,w) and air temperature was measured at 10 Hz using an ultra sonic anemometer-thermometer (HS-50, Gill Instruments Ltd., UK, hereafter termed sonic anemometer) mounted on a horizontal arm at 2 m above ground. Each system was equipped with one of those anemometers. The micrometeorological parameters friction velocity ($u^*$, m s$^{-1}$), roughness length ($z_o$, m) and the Obukhov length (L, m) were computed from the 30 min processed eddy covariance data of the sonic anemometer. Further weather parameters (e.g. global radiation, precipitation) were measured with a standard automated weather station (Campell Scientific Ltd., UK) installed at system M next to the sonic anemometer.

### 2.2.3 Data filtering

The raw MD concentrations were filtered based on the level of light reaching the spectrometer. This led to a data loss between about 1 % and 4 % for the different MD. An additional filter was applied to account for conditions with low turbulence by $u_*$ filtering. As the measurement site is located at the Swiss western plateau which is known for low wind speeds especially during the night a $u_*$ threshold of 0.05 m s$^{-1}$ was applied in order to keep as much data as possible. Flesch et al. (2014) stated that using a $u_*$ value of 0.05 m s$^{-1}$ can be accepted as the data quality does not increase too much by applying higher $u_*$ values. The wind sectors facing towards the farm buildings north and south of the fields were removed as well due to unwanted advection from the nearby farm buildings (see Fig. 1 and Fig. 2). Filtering for $u_*$ and wind direction decreased the data by about 44 % and 49 % for system M and G, respectively.



### 2.2.4 Emission calculation based on dispersion modelling

The emissions were calculated based on dispersion modelling and measurements of $NH_3$ concentrations upwind and downwind of an emitting source. An open–source version of the bLS model by Häni (2017) (based on Flesch et al., 2004) programmed in the statistical software R (R Core Team, 2016) was used to relate the measured 30 minute concentration difference to the
5  unknown emission rate E of the investigated paddocks (see Eq. 1). The coefficient $D$ was determined based on the simulated movement of many thousand fluid particles released at the location of the concentration sensor line and tracked backwards in time. Potential touchdowns inside the specified source area contribute to the magnitude of D.

$$E = \frac{c_{\text{Downwind}} - c_{\text{Upwind}}}{D} \tag{1}$$

The bLS model used wind and turbulence information measured by the sonic anemometer. In order to calculate a concentration footprint for each 30 minute period, averaged data of the wind direction, the standard deviations of the wind components, the friction velocity and values representing the surface roughness were used. Additional geometric information of the source area locations and extensions and the position and height of the miniDOAS measurement paths were provided as well. An intrinsic
assumption of the bLS model approach is that the defined emitting area(s) is homogenous concerning its surface roughness (i.e. vegetation cover) and the source strength. Thus it is assumed that the monitored pasture paddocks are homogenously grazed and the urine and dung patches, representing the main $NH_3$ emission sources, are more or less uniformly (or randomly) distributed on the paddock area.

### 2.2.5 Artificial release experiment

In order to study the uncertainty of the calculated emissions an additional experiment with an artificial gas release was conducted in June/July 2017 next to the sonic anemometer of system M. The source consisted of a grid of 14 critical orifices (100µm diameter, stainless steel, LenoxLaser, USA) which were installed on ground facing upward with a distance of each other of 2 m. The center of the line was connected to a distribution unit which regulated the gas flow with a mass flow controller (red-y smart controller, Voegtlin Instruments, Switzerland). The flow rate, pressure within the grid and the accumulated gas
flow was saved to a hard disk within the housing of the distribution unit. A gas mixture with 5 % $NH_3$ in 95 % $CH_4$ (CarbaGas, Switzerland) was used with a release rate of about 5.5 standard L min$^{-1}$. Two miniDOAS systems (MD2 and MD5) were installed in parallel roughly 6 m north east and south west of the source line to account for the predominant wind directions. Both instruments were installed at a height of about 0.6 m due to the close distance to the artificial source.

### 2.3 Cow and excreta distribution monitoring

As mentioned in chapter 2.2.4 the bLS approach assumes a homogenous distribution of the emission source. On a pasture this is not necessarily the case and can lead to error prone emission estimates (Bell et al., 2017, Laubach et al., 2013a). In order to





assess the spatial distribution of the cow excreta as main emission sources in our experiment, we used two different approaches. The position of dung patches was determined with a hand held GPS device on three rotations on the paddocks X.11 and X.12. In addition, the cow position on the pasture was monitored with a day–night digital camera system and a temporal resolution of 10 minutes. The location of the individual cows were manually marked in a post processing step. However, the night mode

often did not yield useful information and therefore images showing the cow positions during night time were very sparse. The combined information of the dung patch GPS location and the cow position retrieved from the camera images were used to correct the emission estimates for inhomogeneity in the emission source.

## 2.4 Estimation of N excretion on the pasture

The ammonia emission flux, quantified as described above, is a pasture area related quantity. In order to allow a comparison

of the results of the present study with literature reports and with emission inventory models, emission factors were derived by relating the measured emissions to the urine N input from the cows. As N input to the pasture cannot easily be measured the total N and urine N of the excretions of the cows were estimated with a dairy cow nitrogen budget model based on the official Swiss feeding recommendation for dairy cows (Bracher et al., 2011). Input to the model were information concerning the milk yield and N content, the weight of the cows, the calving date, and the crude protein proportional to the N content in

the forage (see Table 2). Milk yield and body weight was measured for each cow on a daily basis whereas data on grass protein was only collected and analyzed eight times between end of April and end of September, but usually close in time to the measurement period. The grass parameters of the systems M and G were averaged for further processing. Crude protein of the maize silage was analyzed three times (beginning of May, mid of July, beginning of September). Missing data were linearly interpolated between the measured values. The N in the excretions were finally calculated as a balance between the N input of

the feed, N storage due to body weight gain and N in milk and excreta for each cow and each day of the year. The break-down in urine N and dung N is based on N balance studies (Bracher et al., 2011). Finally, based on the grazing duration the urine N input to the investigated paddocks was computed. The associated uncertainty of 15 % was estimate by comparing the N budget model to published results of Swiss N excretion studies (Bretscher, unpublished data).

## 3 Results and Discussion

This chapter is organized as follows. The first section (Sect. 3.1) shows the observed ammonia concentrations during the grazing campaign, whereas the next sections present and discuss the emission fluxes. Sect. 3.2 describes the measured area-related fluxes including interference correction and the gap filling leading to cumulative emissions over individual grazing events. The corresponding emission uncertainty and its sources are discussed in Sect. 3.3. The area related emission were converted to animal related emissions using cow and dung distribution monitoring results (Sect. 3.4) and further converted to

emission factors related to animal urine N (Sect. 3.5). In the final section of the chapter (Sect. 3.6) the advantages and problems of the experimental design are highlighted.





## 3.1 Ammonia concentrations during grazing season

The ammonia concentration values observed during the entire measurement campaign had a strong temporal and spatial variability. They were typically in the range of 4-15 µg $NH_3$ $m^{-3}$ with maximum values of about 100 µg $NH_3$ $m^{-3}$. As shown in

Fig. 2 the highest concentrations usually resulted from advection from the nearby farm located in the northern direction of the miniDOAS instruments. This advection is weaker at the southern system G due to the larger distance to the farm. The general concentration pattern is nevertheless very similar for both systems. The highest wind speeds (above 4 m $s^{-1}$) usually resulted in low $NH_3$ concentrations due to a good mixing of the atmospheric boundary layer with lowest concentrations coming from the south–western direction. The higher background concentration from the north–easterly direction is probably a result of a

nearby piggery some 350 m away. During the whole measurement period (beginning of May – mid of October) the MD instruments were online between 62 % (MD 6) and 85 % (MD 2) of the time. Power failure and instrument errors were the main reasons for the partial data loss. The measurement campaign at the Chaumont mountain site (see Sect. 2.2.1) led to a data loss for the first three days during rotation four. During rotation one no data of the MD instruments MD1 and MD6 could be acquired due to instrument errors.

Figure 3 shows an example of concentration measurements of MD2 and MD5 during the first rotation at the beginning of May. During the grazing period on the paddocks X.11 and X.12 the $NH_3$ concentration difference increased due to increased excreta on the field, mainly in the form of urine. Typical concentration differences in the range of about $0 - 8$ µg $NH_3$ $m^{-3}$ for system M and of about $0 - 15$ µg $NH_3$ $m^{-3}$ for system G were measured. A few hours after grazing the concentration differences started to decrease significantly. Typically within the first three to five days after end of grazing (EOG) the concentration differences

reached values around the accuracy limit of the MD devices (about 0.2 µg $NH_3$ $m^{-3}$). Typically for the Swiss western plateau wind speed had a strong diurnal pattern with low wind speeds during night time conditions. This often led to a weak mixing in the boundary layer and subsequent high concentration measurements. In order to avoid error prone emission estimates the concentration values were filtered according to Sect. 2.2.3. This led to low data availability for emission calculation especially during night – time conditions. Precipitation events typically resulted in low concentrations and subsequent low concentration

differences.

## 3.2 Field scale fluxes

The field scale fluxes were based on the concentration differences of the paired MD systems and the computed concentration footprint of the bLS dispersion model. The emissions typically showed a diurnal emission pattern with highest emission values occurring between midday and late afternoon which correlated well with atmospheric driving parameters like air temperature,

wind speed and global radiation (see Fig. 4). The emissions generally increased during the grazing phase (typically grazing duration: 50-70 hours, see Table 1) with a fast subsequent decrease afterwards.

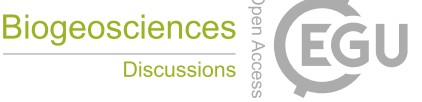



Due to the strong data filtering and unavailable concentration data the emissions have been corrected for missing data values in order to calculate cumulative emissions. The missing values were gap filled by applying a linear interpolation if the time period between available data was less than three hours. Otherwise the values of the standard emission curves (Fig. 5) for the two systems were used. These curves were fitted to the six-hourly averaged daytime emissions of all rotations on paddock

X.11. For this evaluation only half-hourly fluxes without influence from emitting upwind paddocks were used. Because of the low amount of available night time data, it was assumed that the general pattern (increase during grazing period, decrease afterwards) is similar during day and night, and the night time standard emission curves were estimated based on the overall ratio (= 2.54) between the mean daytime and night time emissions during grazing. Due to the limited amount of measured data and the considerable number of possible environmental driving parameters (air temperature, global radiation, wind speed,

precipitation, soil / leaf humidity, see Fig. 4, also Bell et al., 2017; Häni et al., 2016; Laubach et al., 2013b) the emissions were not parameterised as a function of these parameter but only as a function of grazing duration and elapsed time since start/end of grazing. Nevertheless, a good agreement was found using a linear increase of emissions during the grazing period and an exponential decrease afterwards. The decay or e-folding time of the exponential function was evaluated as 28 hours and 23 hours (37 % of maximum value at the beginning) for the systems M and G, respectively. For gap filling of missing flux

measurement values the default curve was fitted to the measurements of each rotation during the grazing phase. This allowed to account for different weather and soil effects during the single rotations.

The applied emission measurement approach as described in Section 2.2 assumes a spatially limited emission between the two measurement paths and negligible emission upwind of the system. However, upwind paddocks were grazed while the measurement paddocks were in the emission decay phase. In some cases, depending on wind direction, the emission sources

on the upwind paddocks can lead to a stronger concentration signal of the inflow compared to the outflow instrument. They interfere with the concentration signals of the paddock(s) of interest and can lead to an underestimation of the true emission. In the strict sense this is a problem of an under–determined systems when less concentration measurement devices are available compared to the emission sources (see Bell et al., 2017). The default emission curves in Fig. 5 were therefore used to estimate the relative influence of grazed upwind paddocks. The estimated difference (see Eq. 2) in concentrations upwind and downwind

$\Delta C_{UD}$ due to upwind grazing can be calculated from the assumed default emission $E_{\text{def}}(t_i)$ of each upwind paddock $i$ and the corresponding bLS coefficients for both MD systems $D_{i,\text{Upwind}}$ and $D_{i,\text{Downwind}}$.

$$\Delta C_{UD} = \sum_i E_{\text{def}}(t_i) \bullet (D_{i,\text{Upwind}} - D_{i,\text{Downwind}}) \qquad\qquad (2)$$

Typically the observed emissions decreased to insignificant values within 3–5 days after EOG. The measured fluxes during the campaign were within a range of 0 to 2.1 µg N-NH$_3$ m$^{-2}$ s$^{-1}$ for system M and 0 to 2.3 µg N-NH$_3$ m$^{-2}$ s$^{-1}$ for system G. The cumulative emissions were calculated based on the gap-filled half hourly fluxes and the area of the individual paddocks (see Fig. 6). Depending on atmospheric driving parameters (mainly precipitation) about 60-70 % of the overall emission occurred during the grazing phase. Precipitation events during that time period led to a significant reduction in emissions with



subsequent higher emission later on (observable especially during rotations two and the higher fluxes on the 14[th] of May in Fig. 6).

Over the entire grazing season, cumulative emissions for the different rotations were retrieved under variable weather conditions with highest air temperatures recorded during rotation three to rotation six and the highest precipitation amounts occurring at the first three rotations (see Table 3). The highest emissions occurred usually at the southern paddock and showed a strong temporal variability depending mainly on the grazing duration and N input. The emissions during rotation seven on system G showed the largest magnitude of all single rotations and fields. This is also in line with the highest N input to the pasture from cow excreta.

### 3.3 Uncertainty of emission flux measurements

### 3.3.1 Effect of different error sources

The performance of the miniDOAS devices for concentration measurements was optimised by adjusting the offsets among all four instruments during the 7-day inter-calibration at the Chaumont site between rotation 3 and 4. During that period the instruments were running in parallel and the measured concentrations (mostly $0 - 2$ µg $NH_3$ m$^{-3}$) were compared to the measurements of wet chemical impingers. It was found that the potential bias between the instruments was below 0.2 µg $NH_3$ m$^{-3}$ and was therefore similar to the results by Sintermann et al. (2016).

Missing flux data was replaced either by values of the standard emission curve (Fig. 5) or by applying a liner interpolation between measurements. The standard emission curves were also used to estimate unwanted interferences in the measured concentration differences from emitting upwind paddocks. In order to test the sensitivity of the emission result to uncertainties in the gap filling method and interferences from upwind grazing, we varied the values of the standard emission curve to 50 % and 150 % of the default values. The sensitivity towards the exponential decay time of the standard emission curve was tested with a systematic increase in the decay time of about 50 % (decay_slow) and a reduction of 30 % (decay_fast). We found (see Fig. 7) that the relative effect of all simulated errors on the cumulative emissions was generally below 20 % for individual rotations (except for few outliers). The highest impact on the emission results was due to the uncertainty in the gap filling of missing values that predominantly occurred during night. Since the simulated error sources are independent, they were combined to an overall measurement related error of 17 % by Gaussian error propagation.

The bLS dispersion modelling is a well-defined approach and was evaluated extensively by Flesch et al. (2005), Harper et al. (2010), and McGinn et al. (2009) who found that the model uncertainty is typically in the order of 20 %. Combining the 20 % uncertainty for the bLS modelling and the 17 % measurement related uncertainty results in total mean systemic uncertainty of 26 %.



### 3.3.2 Artificial gas release

For an overall test of the performance of the applied methodology, artificial gas releases were conducted at the same site in the year after the main experiment in June and July 2017. The gas was only released during stable and westerly winds in order to avoid advection from the nearby barn. Table 4 lists the main meteorological and technical aspects of the individual releases

and shows the corresponding results. The duration of the releases strongly depended on the observed wind speed and varied therefore significantly.

Due to the westerly winds MD 2 detected the upwind concentrations and MD5 the downwind concentrations. All measurements were averaged to 30 minute values and the emissions were calculated following Eq. 1 (see Fig. 8). In order to check the mass flow controller of the artificial source, the release rate of all single orifices were measured during three releases

(release 2, 4 and 5). The observed differences between the summed orifice release rates and the measured mass flow from the gas cylinder varied between -7 % and 9 % with an overall average of only 1 %.

The quality of the calculated emission for each source experiment is defined as recovery rate which is calculated as the ratio of the measured cumulative emissions of the bLS and the cumulative measured emission from the flow controller (see Table 4). Four out of five releases resulted in a recovery rate above 100 % and four release experiments showed a recovery rate

between 88 % and 124 %. Release number one had an exceptional high recovery rate of about 150 %. During that particular release the pressure at the beginning was higher compared to the following ones and might have led to a system which was not fully airtight anymore. The overall mean of 111 % and the standard deviation of 18 % was calculated based on all individual half hourly measurements. This finding is in line with the average estimated systematic error of 26 % (Sect. 3.3.1) indicating that all important error sources have been included and that the existence of an unknown major error source is unlikely.

### 3.4 Animal related emissions

As the bLS approach assumes a homogenous distribution of emission sources within the investigated paddock, the actual distribution of the cow excreta could have a significant influence on the calculated emissions. In order to check the distribution of the excreta on the field GPS monitoring of dung patches and cow monitoring with a camera system was applied. The GPS coordinates were usually acquired within the first 3–5 days after the grazing period on the paddocks X.11 and X.12. However,

dung observations with GPS are only available of two rotations for the paddock M.11, three rotations for G.11 and two rotations for X.12 (see Fig. 9). Day time camera observation for system M was available for the whole measurement campaign. For system G cow monitoring was possible from rotation three onwards.

In order to account for inhomogeneity of excreta on the field, the paddocks were divided into three sections for cow and dung monitoring. The middle section was attributed to the main measurement source area between the MD systems. Figure 10 shows

the relative deviation of the dung and cow density in the main measurement section (between the MD instruments) from the mean density of the entire paddock area. If the cows or dung patches would have been homogenously distributed the relative deviation should be zero. However, a considerable heterogeneous distribution was found for the different rotations and



paddocks. As cow excreta (mainly in form of urine) is the main source of ammonia emissions, missing dung density values were estimated based on a regression analyses ($R^2 = 0.98$) between available parallel measurements of dung patches and cow positions. On the southern pasture (system G) a generally higher proportion of excreta was found between the MD devices in comparison to the averaged field. On the northern pasture (system M) the effect was much more variable with negative

deviations till rotation 5 and positive deviations towards the end of the grazing season.

There is some uncertainty associated to the visual identification (for GPS localisation) of dung patches due to potential double counting or overlooking of dung patches on the paddock, and due to the use of the linear relationship between cow and dung density. But these errors are assumed to behave random-like and are thus relatively small resulting in combined relative uncertainty of about 7 %. This is much smaller compared to the systematic uncertainty of the measured fluxes (see Sect. 3.3.1).

Since there was no cow nor dung monitoring data available for system G during rotation 2, no correction for inhomogeneous excreta density was applied in this case, but a higher uncertainty (25 %) based on the variability of the dung density of the other rotations (Fig. 10).

In order to calculate the emissions per cow and the emission factor for the individual rotations, the derived cumulative emissions were corrected for excreta inhomogeneity by applying the derived relative density values shown in Fig. 10. The

measured emissions per cow and grazing hour (h) stayed rather constant with a value of about $0.64 \pm 0.11$ g N-NH$_3$ cow$^{-1}$ h$^{-1}$ (mean $\pm$ one standard deviation) for system M and about $1.07 \pm 0.12$ g N-NH$_3$ cow$^{-1}$ h$^{-1}$ for system G (see Fig. 11). For comparison, the application of a 10 % standard emission factor for NH$_3$ (EMEP/EEA, 2016) results in larger mean values and a larger variability (system M: $0.99 \pm 0.24$ g N-NH$_3$ cow$^{-1}$ h$^{-1}$; system G: $1.22 \pm 0.31$ g N-NH$_3$ cow$^{-1}$ h$^{-1}$).

The error bars in Fig. 11 represent the total error of the absolute emissions. This error is predominantly due to systematic

effects (see Sect. 3.3.1) that are identical (bLS uncertainty) or very similar (gap filling uncertainty) for the two parallel pasture systems. Therefore these systematic errors are not relevant for the comparison of the two systems, for which only the random uncertainty and the instrument bias uncertainty (see Fig. 7) have to be considered. The random uncertainty for the seasonal mean was estimated from the variability between rotations. In combination with the bias uncertainty this results in a significant mean difference between the two systems of $0.43 \pm 0.13$ g N-NH$_3$ cow$^{-1}$ h$^{-1}$, corresponding to a relative reduction effect of the

N-balanced diet compared to the grazing-only diet of 40 %.

## 3.5 Emission factors for the two pasture systems

The EF values for individual rotations in Table 3 are based on the measured cumulative emissions relative to the urine N deposited (excreted) on the two pasture systems for the different rotations. They range within 4.9 % – 11.1 % for system M

and show generally higher values for system G (range 7.2 % – 16 %). The highest EF values were observed during the second rotation. They are mainly driven by the low N content of the grass on pasture resulting in low estimated urine N excretion (see Table 2). The variation in EF is in contrast to the rather stable measured absolute NH$_3$ emissions as shown in Fig. 11. This may indicate that the analysed grass samples are not fully representative for the selective grazing intake of the cows. On the other hand, an exceptionally high value of the measured emission is unlikely, because a rainfall event started during the second



half of the grazing period and lasted almost two days with a precipitation amount of about 40 mm (data not shown). Typically smaller volatilisation of $NH_3$ is expected during such weather periods (Sommer and Olesen, 2000). A delayed onset of the emissions was observed as described in Móring et al. (2016) after the rain event stopped. However, the emissions were small compared to the ones observed during the first grazing day (roughly one third) and were therefore not able to counterbalance

the reduced emissions of the second part of the grazing period.

The annual average pasture EF and its uncertainty was derived from the overall means of $NH_3$ emission and urine N input and resulted in 6.4 ± 2.0 % for system M and 8.7 ± 2.7 % for system G. The uncertainty of about 1/3 mainly stem from the systematic errors discussed in Sect. 3.3.1 and 2.4.  The found mean EFs are ranked towards the lower end of reported values (5 – 26 % of excreted urine N); see e.g. Jarvis et al., 1989, Bussink, 1992, Laubach et al., 2012, 2013b) but are in line with the

results (6 – 9 %) of the recent study by Bell et al. (2017). A single emission factor as used in many inventory models (e.g. EMEP/EEA, 2016; Kupper et al., 2015) would not be able to reflect the observed difference of 2.3 % between the two grazing/feeding systems in our experiment. The reduction in EF for system M is not statistically significant but may indicate a nonlinear effect of the N input rate on the $NH_3$ emission, similar to the findings of the recent literature synthesis study by Jiang et al. (2017) who reported a higher emission factor with increasing fertiliser N application. Thus the optimised N-

balanced feeding strategy may decrease the $NH_3$ emission even more than expected from the reduced urine N excretion.

### 3.6 Advantages and problems of experimental setup

The present field experiment was optimised to measure the $NH_3$ emissions of two neighbouring pastures managed in an intensive rotation. The periodic high density of animals (55-70 cows ha$^{-1}$) and fresh excreta on the grazed paddocks resulted in intermittent high fluxes and allowed to observe the temporal behaviour of the emissions (Fig. 5, Fig. 6). This would not be

possible on a continuous grazing system with much larger paddock sizes and accordingly smaller excreta densities and emissions. For continuous grazing on large fields other micrometeorological measurement techniques like the eddy covariance (Ammann et al., 2012) would be preferable. The small paddock sizes in this study also kept the cow excreta heterogeneity on a moderate level, whereas on larger free range grazing areas the animals often gather at the same place (Cowan et al., 2015) leading to a more complicated quantification of the EF. While the distribution of dung patches and cows was monitored by

means of visual inspection or evaluation of the camera images, a direct localisation of urine patches was not possible in this way. Sensors for urine patch detection exist, but are either still in development (Kumar et al., 2016), relatively expensive (Quin et al., 2016), or unpractically for field scale experiments (Dodd et al., 2015). Therefore we assumed a similar density distribution of dung and urine patches on the paddock analogous to Luo et al. (2017).

The present setup with the parallel pastures and accordingly similar micrometeorological conditions constituted an effective

way to analyse the difference between the two systems as the main systematic uncertainty source of the single pasture emissions (bLS, Sect. 3.3.1) were cancelled out. However, subsequent grazing on neighbouring upwind paddocks could produce interferences with the measurements that could be corrected only in an approximate way. Another error source arose due to the strong variability of the measured crude protein in the grass with consequent high variability of the estimated N in the



urine. It was not directly measured as automated monitoring techniques for urine N on the pasture are not yet mature enough and still have some limitations regarding the animal welfare (Misselbrook et al., 2016). Manual measurements of the urine N amount were outside of the scope of this project due to the laborious work.

## 4 Concluding remarks

In a paired field experiment $NH_3$ emissions on two pasture systems were measured for an entire grazing season under real practice conditions. The herds of the two pastures were kept in an intensive rotational grazing management with different protein to energy ratios resulting in different N excretion rates. The fast rotation with a short but high stocking rate and excreta deposition within the grazed paddock allowed to observe the temporal dynamics of the corresponding $NH_3$ emission. Maximum emissions were found at the end of each grazing phase on the investigated area. Afterwards an exponential decay of the

emissions led to non-significant low values typically within 3-5 days. A diurnal emission pattern with peaks during the afternoon was observed on all rotations.

Monitoring of the cow and dung density distribution was essential for a quantitative comparison of the two systems. The emission per cow and grazing hour showed only a very limited variation over the season but a distinct difference (40 %) between the two systems. About half of this difference could be explained by the different urine N excretion rate of the two

herds. The resulting average EFs were $6.4 \pm 2.0$ % and $8.7 \pm 2.7$ % for the herd with the N balanced diet and the herd with the N surplus in the forage, respectively. Thus the experiment showed the large potential of an optimised feeding strategy to reduce $NH_3$ emissions. The results can also serve as a validation for the Swiss national emission inventory for $NH_3$ emissions on pastures. It is recommended for further studies to include the regular analyses of the N content in the urine in order to overcome the associated uncertainties.

*Data availability.* Data obtained in this study will be online available at the time of publication from the data repository zenodo.org. (the DOI will be included in the final paper version).

*Competing interests.* The authors declare that they have no conflict of interest.

*Acknowledgements.* We gratefully acknowledge the funding from the Swiss National Science Foundation (Project NICEGRAS, number 155964). We wish to thank Lukas Eggerschwiler, Robin Giger, Walter Glauser, Harald Menzi, Andreas Münger and Jens Leifeld for support in the field and helpful discussions. We especially acknowledge the contribution of Harald Menzi in the design and planning of the experiment. We are grateful to Albrecht Neftel for the helpful discussions and advise

30  concerning the MiniDOAS measurements. We thank Daniel Bretscher for the support with the N balance computation of the cows and the discussions of these data. This work was supported by a MICMoR Fellowship through KIT/IMK-IFU to Karl Voglmeier.



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





**Table 1: Summary of grazing rotations 2016 on subplots X.11 and X.12 investigated for NH₃ emissions**

| Grazing rotation | Start date of grazing | Grazing duration [h] | Time in barn during grazing [h] |
|:---:|:---:|:---:|:---:|
| 1 | 2016–05–09 | 44.5 | 11 |
| 2 | 2016–05–26 | 46.5 | 9 |
| 3 | 2016–07–04 | 37 | 8.5 |
| 4 | 2016–07–26 | 51 | 20.5 |
| 5 | 2016–08–10 | 29 | 8 |
| 6 | 2016–09–04 | 36.5 | 17 |
| 7 | 2016–09–26 | 55 | 13 |



**Table 2: Measured driving parameters and resulting urine N and feces N of the animal N budget model averaged for the individual rotations and for each herd (system M / G). If only one number is given it corresponds to both herds simultaneously. Rotation 4 is not shown due to missing miniDOAS measurements.**

| Rotation | 1 | 2 | 3 | 5 | 6 | 7 |
|---|---|---|---|---|---|---|
| Animal weight (kg) | 639 / 635 | 646 / 635 | 636 / 637 | 630 / 630 | 630 / 637 | 633 / 637 |
| Days since calving | 187 / 199 | 204 / 216 | 182 / 197 | 217 / 218 | 242 / 243 | 258 / 265 |
| Milk yield ($kg\ cow^{-1}\ day^{-1}$) | 26.7 / 25.3 | 24.4 / 23.7 | 25.0 / 23.8 | 23.3 / 23.3 | 23.2 / 20.6 | 19.2 / 15.9 |
| Grass crude protein ($g\ kg\text{-}DM^{-1}$) | 203 | 147 | 178 | 200 | 218 | 200 |
| Maize crude protein ($g\ kg\text{-}DM^{-1}$) | 91 / – | 91 / – | 89 / – | 80 / – | 72 / – | 71 / – |
| Urine N ($g\ cow^{-1}\ day^{-1}$) | 274 / 324 | 135 / 157 | 218 / 269 | 266 / 326 | 295 / 371 | 244 / 317 |
| Feces N ($g\ cow^{-1}\ day^{-1}$) | 160 / 157 | 146 / 146 | 150 / 152 | 150 / 151 | 153 / 149 | 147 / 142 |



**Table 3: Cumulative emission results for paddocks X.11 and X.12 (combined) of the two pasture systems (M / G) during the individual rotations. Corresponding averaged weather parameters and N excretion input to the paddocks are also listed. Rotation 4 is not shown due to missing miniDOAS data at the beginning of the rotation.**

| Rotation | 1 | 2 | 3 | 5 | 6 | 7 |
|---|---|---|---|---|---|---|
| flux data coverage (until 3 days after EOG) [%] | 55 / – | 65 / 44 | 34 / 39 | – / 30 | 50 / – | 51 / 50 |
| Air temperature [°C] | 11.9 | 14.8 | 18.9 | 17.8 | 18.08 | 14.37 |
| $u_*$ [m s$^{-1}$] | 0.13 | 0.15 | 0.12 | 0.09 | 0.11 | 0.13 |
| Precipitation [mm] | 51.2 | 75.4 | 60.6 | 6.5 | 33.3 | 9.7 |
| Cumulative emission [g N-NH$_3$] | 332 / – | 349 / 600 | 357 / 496 | – / 341 | 277 / – | 330 / 726 |
| N excretion total [kg] | 9.6 / 10.7 | 6.5 / 7.1 | 6.8 / 7.8 | 5.9 / 6.9 | 8.2 / 9.5 | 10.8 / 12.6 |
| N excretion urine [kg] | 6.1 / 7.2 | 3.1 / 3.6 | 4.0 / 5.0 | 3.8 / 4.7 | 5.4 / 6.7 | 6.7 / 8.7 |
| EF relative to urine N input [%] | 5.5 / – | 11.1 / 16.4 | 8.8 / 10.0 | – / 7.2 | 5.1 / – | 4.9 / 8.3 |





**Table 4: Environmental conditions and recovery rates during the individual artificial gas release experiments. Averaged values during the release periods are shown.**

| Date | Release duration [hour] | Friction velocity [ms$^{-1}$] | Radiation [Wm$^{-2}$] | Wind direction [°] | Air temperature [°C] | Recovery rate [%] |
|---|---|---|---|---|---|---|
| 09–06–2017 | 1.5 | 0.18 | 451 | 269 | 20.1 | 150 |
| 12–06–2017 | 2.5 | 0.26 | 628 | 272 | 25.6 | 124 |
| 19–06–2017 | 3.5 | 0.25 | 915 | 256 | 26.0 | 88 |
| 27–06–2017 | 1.5 | 0.26 | 891 | 230 | 24.6 | 114 |
| 12–07–2017 | 3.0 | 0.53 | 756 | 240 | 24.1 | 112 |



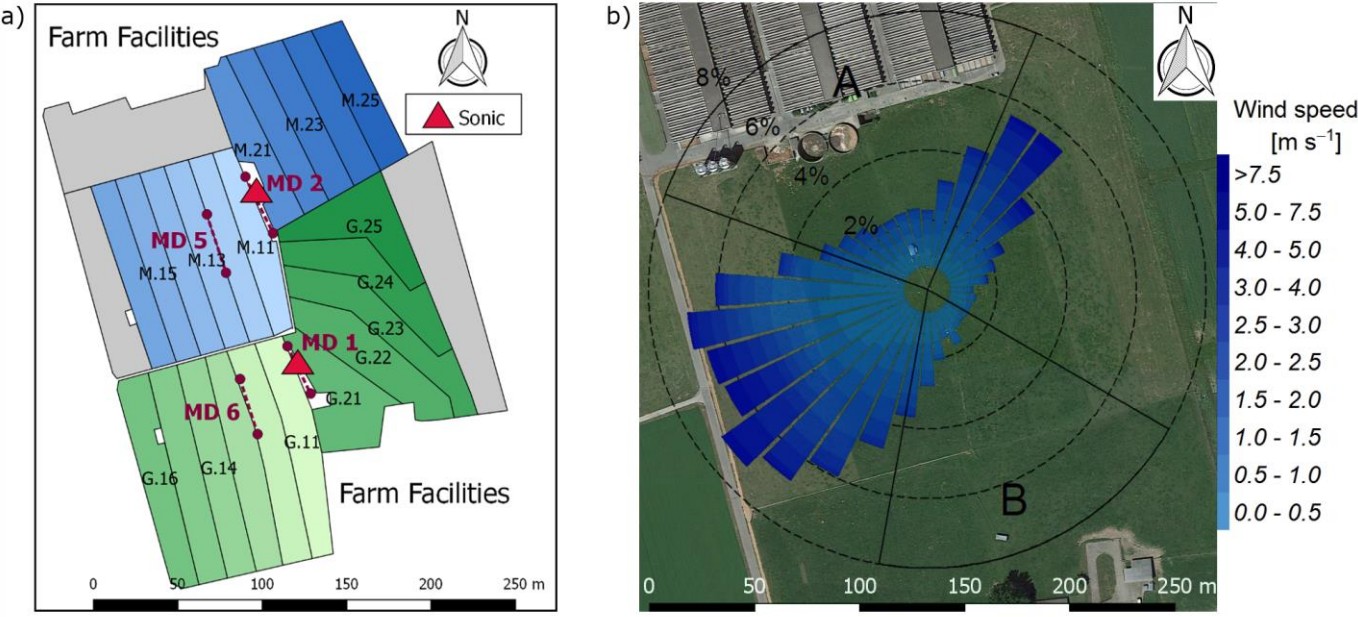

**Figure 1: a) Measurement site with the pastures for the two herds (blue: grass diet with additional maize silage; green: full grazing regime; grey: optional pasture areas) and the division into the paddocks (M.11-M.25, G.11–G.25). Additionally the location of the two sonic anemometers and the four miniDOAS systems (MD1 – MD6, naming based on serial number) are shown. b) Wind distribution for the northern sonic anemometer with the corresponding sector contributions (black dotted circles) for the period May – October 2016. The areas A and B indicate wind sectors from which advection from nearby farm building can occur. The wind distribution was overlaid on a Google Earth image of the experimental area (Map data: Google, DigitalGlobe)**





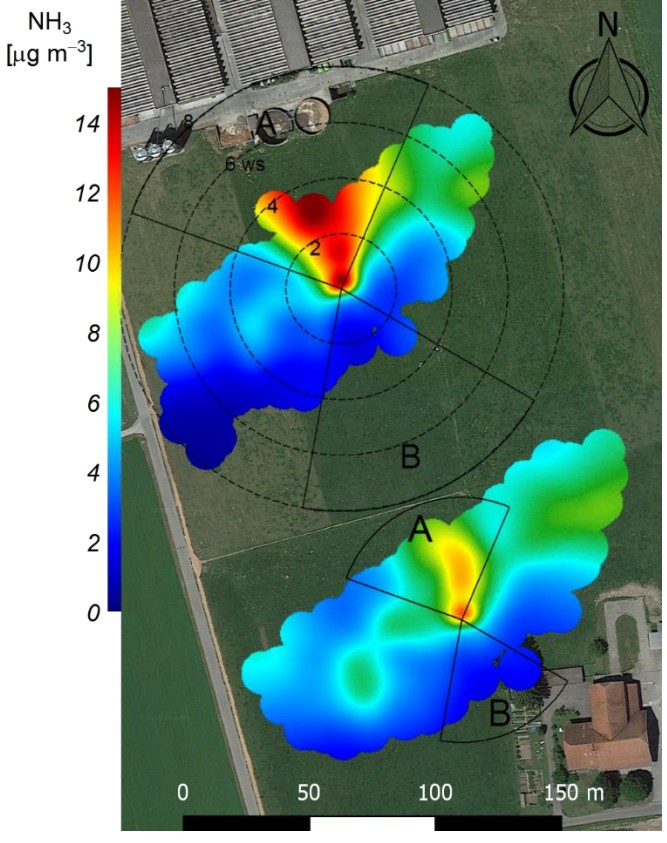

**Figure 2: The Polar plot shows the averaged NH₃ concentration of the miniDOAS MD5 (top) and MD6 (bottom) depending on wind direction and wind speed (black dotted circles) for the period May – October 2016. The sectors A and B indicate areas with either high NH₃ concentration from farm buildings or otherwise unfavourable wind direction due to the measurement setup. The polar plots were produced using the R software package openair (Carslaw and Ropkins, 2012) and overlaid on a Google Earth image of the experimental area (Map data: Google, DigitalGlobe).**





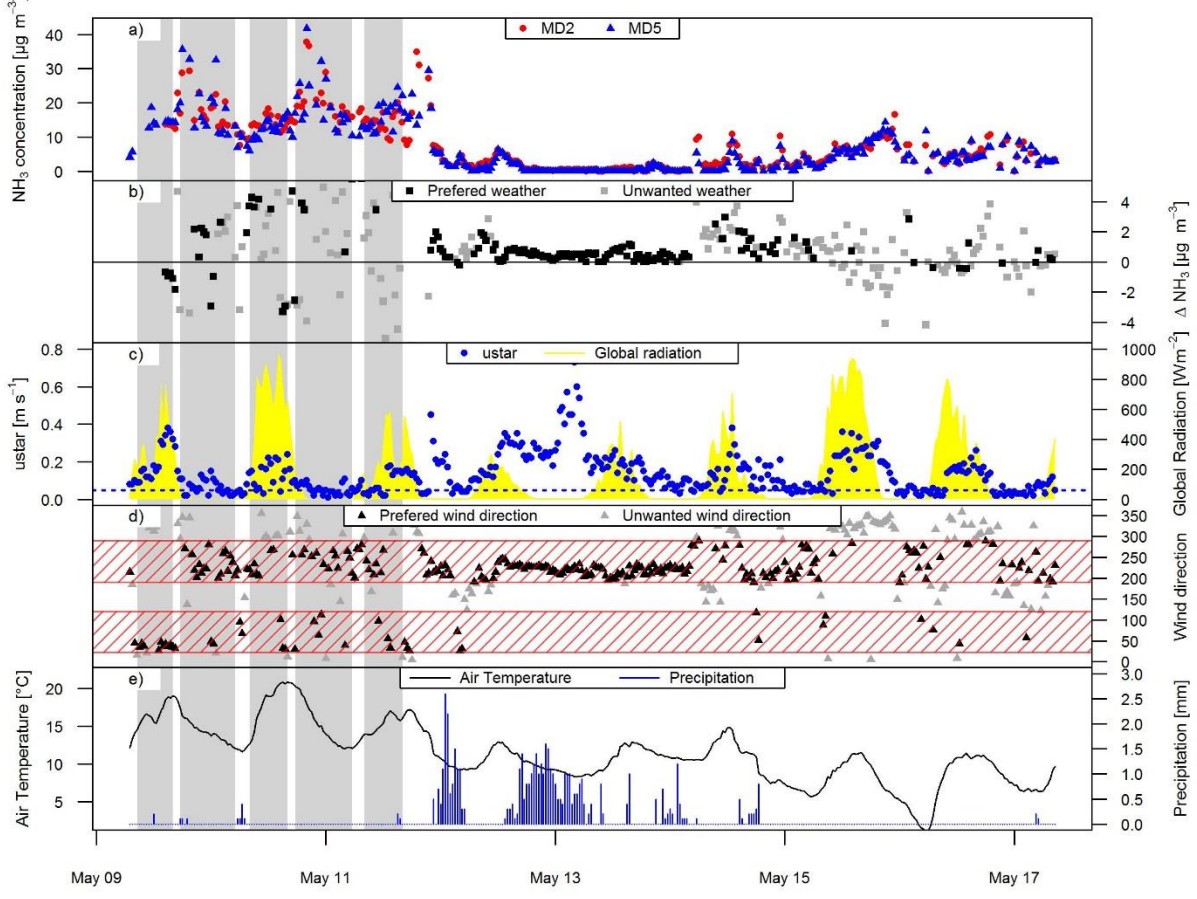

**Figure 3: Time series of a) MD concentration measurements (MD2 and MD5) of system M and b) corresponding difference in concentration. The concentration differences during good wind conditions are shown in black colour while the grey colour indicate concentration differences during undesirable weather conditions. c) Time series of u* and global radiation. The blue dashed line indicate the 0.05 m s⁻¹ u* threshold. d) Time series of wind direction. Wind direction values overlapping with the preferred wind sector (avoiding sector A and B, see Fig. 2) are shown in black colour. The preferred wind sectors are indicated by the red area. e) Time series of air temperature and precipitation. The grey shaded area indicates grazing on the paddocks in between MD2 and MD5.**





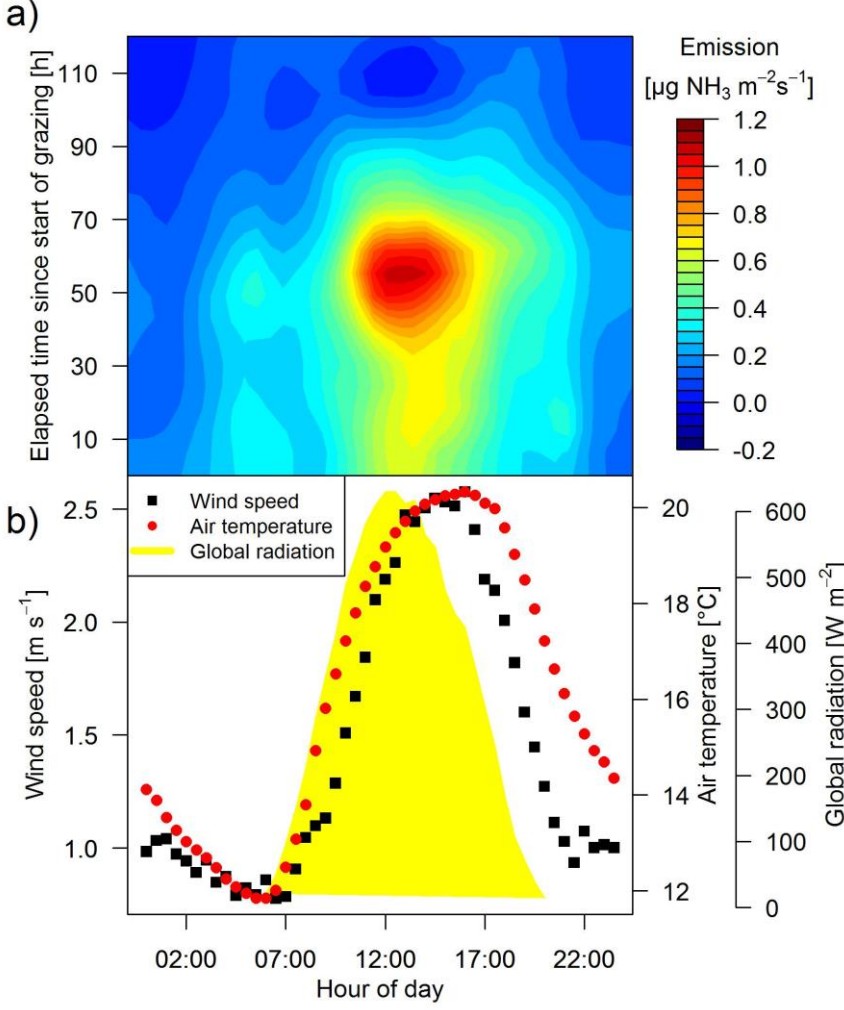

**Figure 4: a) Measured averaged half hourly fluxes of all rotations of the system M depending on hour of day and elapsed time since grazing on the paddocks in between MD2 and MD5 started. b) Half hourly averaged values of global radiation, wind speed and air temperature measured at system M during May to October 2016.**


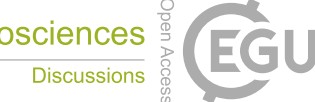

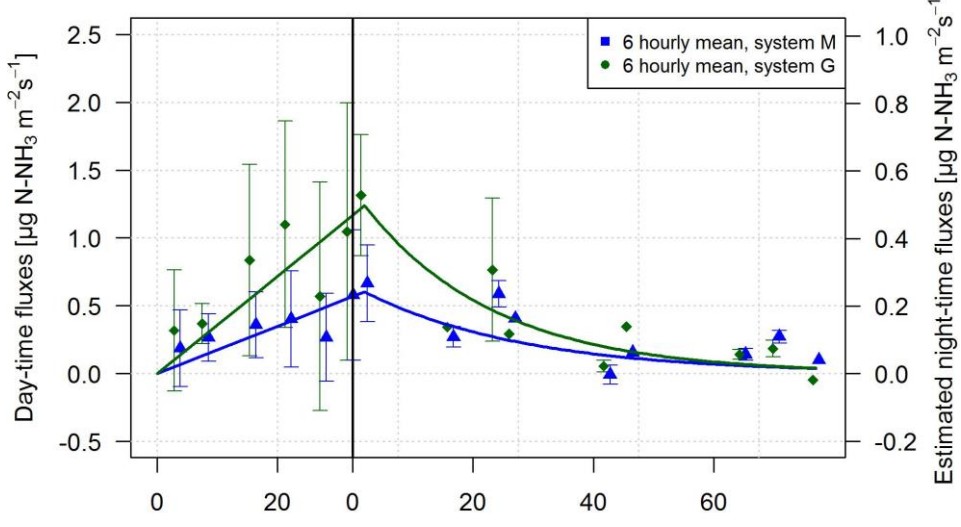

**Figure 5: Default emission curves for system M (blue) and system G (green) based on the grazing duration. Linear increase from start of grazing until three hours after end of grazing and exponential decrease afterwards were fitted to the 6-hourly averaged values of the measured daytime fluxes. Night time emissions were estimated as constant fraction of the daytime emissions (see text). The black vertical line indicates the end of grazing. For better readability the data points for the two systems were slightly shifted horizontally.**




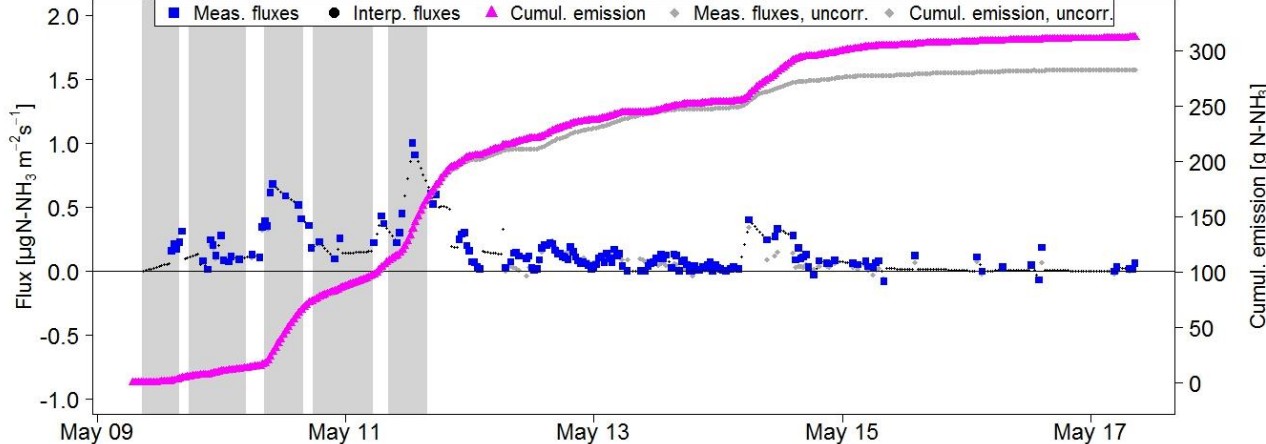

**Figure 6: Measured emission of paddocks M.11 and M.12 (between MD2 and MD5, see Fig. 1a) during rotation 1. Missing half-hourly flux data were filled based on either linear interpolation or on the default emission curve (Fig. 5) in order to compute the cumulative emission. For comparison the uncorrected emissions (interference of upwind grazing acc. to Eq. 2 not considered) are**
5 **also shown. The shaded time intervals indicate grazing on the investigated paddocks.**





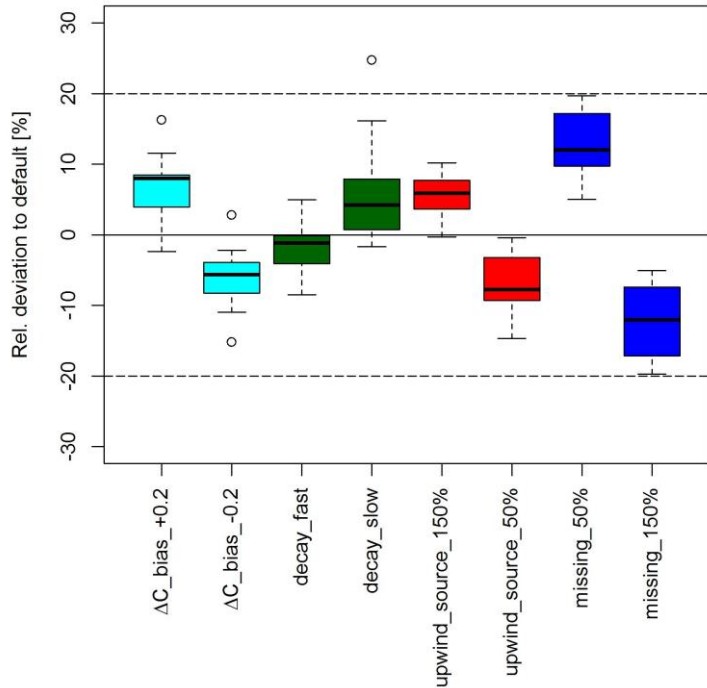

**Figure 7: Sensitivity analysis of various error sources on emission results for individual rotations. Each boxplot shows the resulting relative effect of a potential systematic error. The investigated effects include the over- or underestimation of: the offset in concentration measurements (cyan), exponential decay times of the standard emission curves in Fig. 5 (green), magnitude of standard emission curves used for upwind source interference correction (red) and for gap filling (blue).**





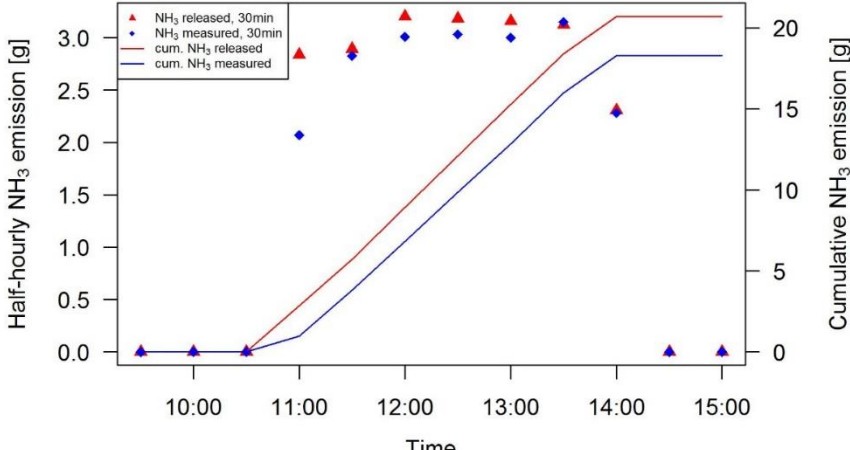

**Figure 8: Released (red) and measured (blue) NH₃ emissions during the artificial source experiment 3 on the 19 June 2017. The measured emissions were quantified using the concentration difference of the miniDOAS systems MD2 and MD5 and the corresponding modelled bLS concentration footprints.**

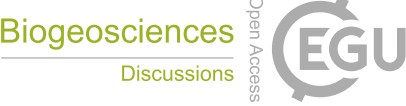

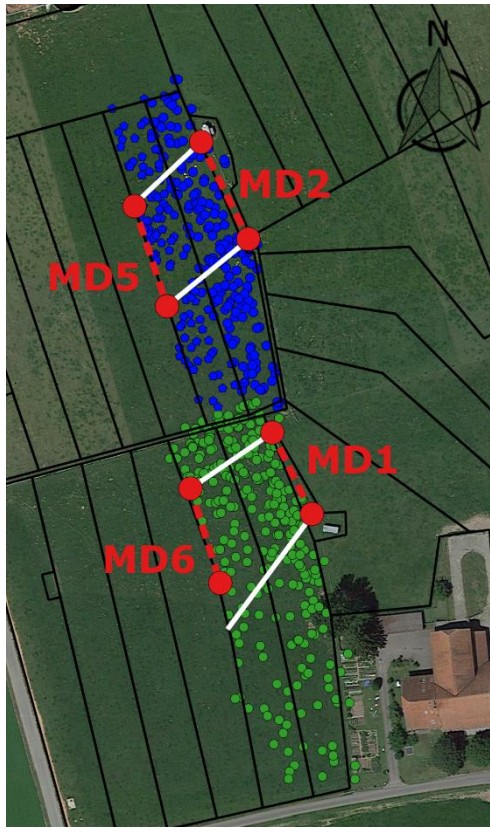

**Figure 9: GPS dung positions overlaid on a Google Earth image of the experimental area (Map data: Google, DigitalGlobe). The positions of the MD devices are indicated by the red dots and the red dotted line show the light path between the MD systems and the light reflector. The white lines show the border between monitoring sections.**





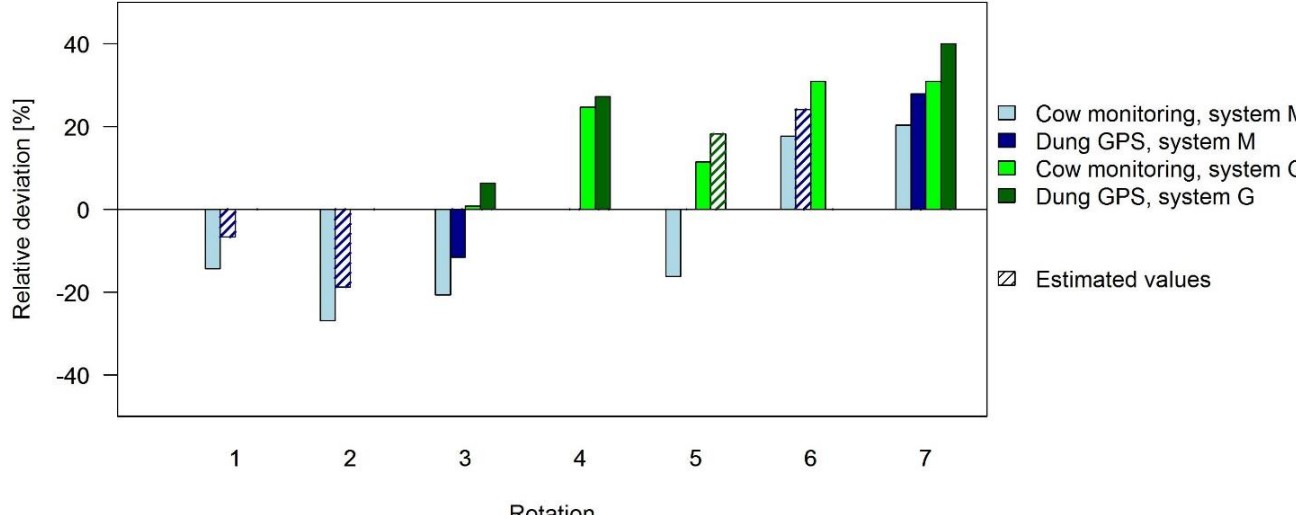

**Figure 10: Relative deviation of observed dung patch and cow density within the measurement area between the paired MD instruments (see Fig. 9) from the whole paddock mean density. For the rotations without dung observations, the corresponding deviations were estimated (hatched bars) from a regression analysis between available dung and cow densities (only for rotations with available emission results, see Table 3).**





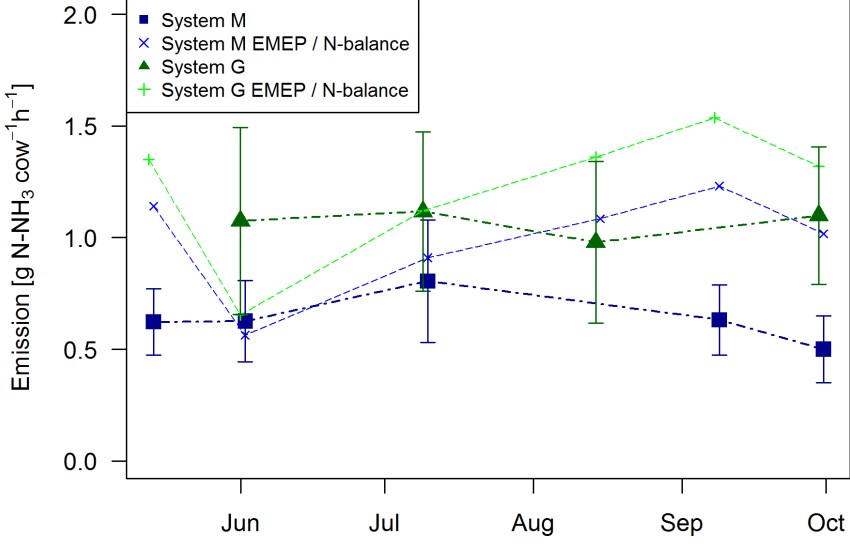

**Figure 11: Emissions per cow and grazing hour for system M and system G. Measured values (thick dots and lines) in comparison to estimated values based on urine N amount from the N balance model and the EMEP standard emission factor for ammonia (10 %, see EMEP/EEA, 2016). The error bars were calculated based on the methodological uncertainty (Sect. 3.3.1) and on excreta density uncertainty (Sect. 3.4).**