# Peer review of "Ammonia emission measurements of an intensively grazed pasture"

_Biogeosciences, 2018_

## Referee Comment (RC1) · Anonymous Referee #1 · 14 Apr 2018

This study investigated ammonia emissions from the urine (and also from dung patches) produced by cattle in a grazing operation. The manuscript needs work as it needs to be more concise, and also needs more informative regarding exactly what was done in the methodology (e.g., modifications to bLS, procedure for correcting fluxes). The weakest part of the study was investigation in estimating urine patch locations to avoid assumptions on the uniformity of the source on pasture (by measuring dung pile locations and position of the cows). In applications where bLS is often used in non-uniform sources, it is realized that the detector should be some distance downwind to minimize the impact of non-uniform source on emissions but close enough to resolve horizontal gradient (elevated background concentration, a possible problem in this study). The strongest part of the study was the N balance of the two pasture

systems and the possible mitigation of ammonia emissions from pastures with grazing cattle that excrete manure. Line numbering should be continuous , on every line, and the line spacing should be doubled to assist the reviewers in editing the manuscript. Page/line 1/7 delete 'in ambient air' 1/8 replace 'have therefore a large uncertainty' with 'is uncertain' 1/14 replace 'and prevailed within a range of' with 'and ranged from ' 1/15 insert 'maximum of x $\mu$g N-NH3 m-2 s-1 at the end' 1/18 replace with 'Dung and cow location were monitored to account' 2/5 'about eight times lower' - could not find this in Kupper et al (2015) - re-check citation 2/8 delete 'the input of' 2/10 delete 'see' here and throughout text, redundant 2/10 separate authors by semicolon 2/13 abbreviate 'ammonia as 'NH3' throughout MS 2/15 use 'and further developed' 2/18 use '12-d period' 2/25 was the model 'WindTrax' by ThunderBeach Scientific - need to cite model 3/10 what was the topography (slope, barriers to flow, etc) 4/5 use 30-min averages' 4/12 were pressure and temperature corrections needed, if so give calibration factors 4/12 was light intensity used to filter data, if so, give range 4/28 use 'Figs. 1 and 2' 5/3 describe the model, and what modifications were made to Flesch's model, what was different 5/4 use '30-min concentration' - same for 5/12 5/13 abbreviate as 'u*' previously defined 5/15 however, the 'underlying' assumption of homogeneity of the emitting surface is less true with increased distance between the source and detector, please include this - it is unclear why the bLS model was not run in its entirety 25/5 state the given NH3 concentration certification 5/31 'this is not necessarily the case' - this deserves further comment 6/1 top page 11 states urine patches are the most important factor - then two ways of trying to estimate where these patches exit is tried by GPS of the dung piles and by locating the position of the cows - this cannot be direct emission map of ammonia since cows do not necessarily defecate and urinate at the same location, and the position of the cow adds little information to estimate urine patches. 6/7 it is not clear that the error would be reduced by compounding the errors in locating the urine patches, as opposed to assuming a uniform distribution, especially when the uniform criteria declines in importance with some distance downwind. 6/25 no need to introduce upcoming sections, delete lines 25 to 31 7/15 no need to introduce Fig 3 - delete sentence 7/27 need to expand by providing information on what was done regarding the bLS footprint 7/31 is this 50-70 hours per week? 1/8 what is a 'strong' data filter - need to rewrite 8/8 explain where the value '2.54' came from 8/13 use '28 and 23 hours' 8/20 use 'greater concentration signal' 8/17 are you saying that your design, at specific wind directions, caused an interference of the incoming concentration (upwind) measurement which lead to an under-estimate of emissions - why not filter out the estimates? 8/22 do you mean 'fewer concentration detectors' 8/32 use 'half-hourly fluxes' 9/3 use 'recorded' not 'retrieved' 9/4 use 'greatest air temperature' and 9/5 'greater emissions' 9/6 neither grazing duration nor N input is found in Table 3 - where are these data? 9/16 use 'data were' 10/2 'artificial gas' awkward, best to say 'tracer gas' and give the species name, e.g., SF6 10/8 use '30-min values' 10/10 usually as an alternative to mass flow controller, the entire tank is weight before and after, was this done in this study? 10/11 use '-7 and 9%' 10/15 use '88 and 124%' 10/16 do you mean 'air pressure' 10/16 don't understand the set-up, what was no longer air tight - needs clarification, also need to indicate why air pressure is involved in recovery 10/18 use 'half-hourly measurements' 10/19 'an unknown major error source is unlikely' - what does this mean, if unknown how can it be unlikely, delete this sentence as it adds no information - were the results used to correct the emission or was it used to characterize the data? How sure are you that the difference was systematic, if this is important there needs to be a t-test done and if different then an accuracy analysis preformed to break the difference into systematic, random and slope errors 10/25 use 'for two' 11/5 use 'until' 11/11 how was this correction done in all systems except system G rotation2, needs clarification - also need to document what this means for this latter value that was not corrected 11/11 use 'greater uncertainty' 11/30 use '7.2 -16%' same for line 29 12/3 cited reference not listed 13/6 delete 'under real practice conditions'

---

## Referee Comment (RC2) · Anonymous Referee #2 · 5 Jun 2018

This manuscript is an important contribution to the field of ammonia emission evaluation and more widely nitrogen management in agriculture. Indeed, this manuscript presents, evaluates and analyses the results of a field scale methodology for evaluating nitrogen losses as ammonia volatilisation from two intensively grazed pastures in Switzerland. The method is based on inverse Lagrangian dispersion modelling and line integrating open path DOAS instruments. Two cow herds with differing nitrogen diets are compared over small fields ensuring a quite high grazing density hence insuring a good sensitivity of the method. The uncertainties of the methodology are evaluated and the homogeneity of the dung patches is assessed based on camera monitoring and observations. The inverse dispersion methodology is also evaluated against a controlled ammonia source of similar dimension as the fields and proves to work quite well.

[Figure]

This study convincingly demonstrates that N-rich diets lead to larger emissions than N-balanced diet (about 30% larger). The methodologies are scientifically sound and the methods and results are well presented, although some clarifications are needed both on the methodology and presentations (see comments below). The figures and tables are clear, and the manuscript is well written. The topic is of great interest for both the scientific community and decision makers.

General comments

* The methodology used for gap-filling emissions during night-time (and to a lesser extent for wind sectors coming from the surrounding farms) may be questionable. Indeed, the authors assume that night-time fluxes may be gap-filled based on day-time fluxes, but ammonia fluxes are fundamentally based on thermodynamical equilibrium at the surface (gas-liquid and acid-base equilibriums). This means that (1) the surface ammonia concentration is exponentially increasing with surface temperature due to the gas-liquid or Henry equilibrium, and we would hence expect lower emission at night due to lower temperature and hence lower concentration at the surface; (2) similarly, since ammonia fluxes are proportional to a concentration difference between the surface and a reference level, lower turbulent exchanges at night are expected to decrease night-time ammonia emissions. This means that using daytime fluxes amplitude and dynamics may systematically bias the emissions towards higher values. I would recommend a discussion on that point which may include a study on the temperature and u* dependency of the ammonia emissions. Authors could refer to e.g. Flechard et al. (2013) for details on ammonia the points raised above.

* Reference to the work of Moring et al. (2016) is lacking. In reference to this work, I wonder if considering the source as a mosaic of emission and deposition hot-spots rather than a distribution of emission patches would conceptually change the results presented here. Could the author elaborate on this question?

* The authors have made a good job in synthesising complex experiments and methodologies, though there is sometimes a feeling of over-simplification, which makes the overall evaluation of the quality of the results difficult. I would hence recommend adding a supplementary section to provide some details, in particular regarding:

(1) The uncertainty analysis requires more details and especially on the gap-filling of emissions using the standard curve on Figure 5. An example showing a reconstructed emission would be beneficial here. It is difficult also to understand if the uncertainty analysis on gap-filling spans the actual variability in the fluxes shown in Figure 5 (the error-bars in Figure 5 would also need to be explained).

(2) The methodology used to derive the dung patches distribution, the way relative deviation of this dung patches Is calculated and the way it is used to correct cow-based emissions all need clarifications. I think authors should mention previous work on patches emissions by Moring et al. (2016).

(3) The description of the artificial source quality would benefit from more details on the homogeneity of the emissions, the pressure and flow rates stability. Some more examples on measured concentration and retrieved emissions during these trials would also be beneficial as these data were not published previously (to my knowledge).

Detailed comments

* P4-L7: delete "important" before reference spectrum

* P4-L14-20: some details on the meteorological instruments may be useful. Please evaluate also how important may be high frequency losses on u* and H with 10Hz acquisition at 2 m above ground.

* P4-L25-30: 44%-49% missing values for low ustar may actually bias the analysis (see major comments)

* P5-L6: please give number of thousands of trajectory

* P5-L9: please give units of E, C and D. I woud also suggest to explicit the hypotesis

behind this equation: actually Cdown = D*S + Cup, where not other nearby sources are assumed.

* P5-L15-18: are the two hypothesis of a uniform and continuous distribution and of a random uniform distribution of sources strickly identical for inverse dispersion?

* P5-L31: the work from Moring et al. (2016) should be referred to here and after.

* P6-L4-8: the method used correct the cow-based emissions based on images and GPS needs to be detailed, as suggested in the major comment section.

* P8-EQ-2: The meaning of ti is unclear.ïĄĎCUD and Dup and Ddown should also be time dependent. Please clarify. I would suggest rather using t and Edef, i(t) etc.

* P9-L11-15: I woulld suggest showing the concentration inter-comparison figure in a supplementary material.

* P9-L28: please explain what is a "systemic" unncertainty.

* P10-L2: I am not sure the word "stable" is appropriate here as it may be understood as "stabe thermal stratification".

* P10-L7-10: I wonder what is thevariablity in the release rate between the critical orifice. I also wonder if the atmospheric pressure has an influence onf the realese rate at 30 minutes but also over short time scales (seconds). Finally, what is the expected (or even recorded) effect of wind speed variations on release rates : would one expect some ventury effects on the release rates?

* P10-L28-32: It is quite unclear what the "relative deviation of the dung density" really is. I would suggest providing the exact equation.

* P11-L1-3: it is unclear how exactly missing values are obtained from regression analysis. Could the authors elaborate on that?

* P11-L6-12: I would suggest giving details on how the uncertainties are aggregated

(may be an equation in a supplementary section?)

* Table 2: I would suggest finding a way to separate more clearly G and M in this table as in Table 3

* Table 3: I suggest only proving 1 digit for temperature and none for rainfall.

* Figure 5: could you specify the meaning of the error-bars. I would also suggest using negative time values on the left.

* Figure 8: Could you provide error bars on both released and inverse modelling with measurements. I would also suggest changing one

* Figure 10: Please explicit the term "relative deviation" in the legend.

* Figure 11. Please explicit if error bars are standard deviation, standard errors or interquartile.

References

Flechard, C.R., Massad, R.S., Loubet, B., Personne, E., Simpson, D., Bash, J.O., Cooter, E.J., Nemitz, E. and Sutton, M.A., 2013. Advances in understanding, models and parameterizations of biosphere-atmosphere ammonia exchange. Biogeosciences, 10(7): 5183-5225.

Moring, A., Vieno, M., Doherty, R.M., Laubach, J., Taghizadeh-Toosi, A. and Sutton, M.A., 2016. A process-based model for ammonia emission from urine patches, GAG (Generation of Ammonia from Grazing): description and sensitivity analysis. Biogeosciences, 13(6): 1837-1861.

---

## Author Comment (AC1) · 25 Jun 2018

bg-2018-86

**Author response to comments of referee #1**

We'd like to thank reviewer #1 for his answer and appreciate his valuable comments.

(referee comments are printed in *italic*, author responses are printed in blue)

*1. The weakest part of the study was investigation in estimating urine patch locations to avoid assumptions on the uniformity of the source on pasture (by measuring dung pile locations and position of the cows). In applications where bLS is often used in non-uniform sources, it is realized that the detector should be some distance downwind to minimize the impact of non-uniform source on emissions but close enough to resolve horizontal gradient (elevated background concentration, a possible problem in this study).*

We do not really understand this statement. In our view, the estimation of the excreta distribution on a real grazed pasture is, despite the necessary approximations, one of the strength of the present study since this issue is either missing in comparable studies (Bell et al., 2017), artificially forced by distributing urine manually on the pasture (Laubach et al., 2012) or by forcing unrealistically high excreta densities during short experiments (Laubach et al., 2013b). We made dung patch surveys and we applied a robust method to estimate dung patch densities based on visual cow monitoring with camera systems. As pointed out in the manuscript, it is a valid assumption that urine and dung patches are similar distributed on the paddock (P12 L28). Auerswald et al. (2010) also found a similar spatial distribution between urine and dung patches on a low intensity pasture.

We are fully aware that placing a detector further downwind minimizes the impact of a non-uniform source. Nevertheless, as referee #2 also pointed out, the small fields ensured a temporarily high stocking density and hence a good sensitivity of the concentration measurement method. We would loss this advantage if placing the detector further downwind. Additionally, for maintaining two realistic grazing systems over an entire season, it was not possible to keep the animals in smaller paddocks. As we investigated a rotational management, placing the detector further downwind would have resulted in an average emission measurement over multiple paddocks (or differently managed areas). Therefore monitoring the temporal dynamics of the emissions (increase with grazing duration, exp. decrease afterwards) would not have been possible.

However, we realized that the presentation of this issue in the manuscript was not optimal and this may have influenced the referee comment. Therefore we will improve the manuscript in this respect (see specific comments below).

*Detailed comments:*
*For the majority of the minor (mostly language related) comments we follow the referee suggestions. Here only the comments that need an answer are listed.*

*1/15 insert 'maximum of x µg N-NH3 m-2 s-1 at the end'*
As the maximum emissions at the end of the grazing period varied (mostly due to different grazing duration), we would like to keep the sentence unchanged. The overall maximum flux value is included in the range given in the previous sentence.

*2/5 'about eight times lower' - could not find this in Kupper et al (2015) - re-check citation*
The 'eight times lower' factor was calculated based on the TAN flows in Fig. 4b in Kupper et al. (2015). That figure shows that the relative $NH_3$ emission of grazing livestock (8.9% of excreta TAN) is 7.6 times lower compared to indoor housing including storage and spreading of manure (67.8% of excreta TAN). This factor was rounded to 'about eight'. We will make the reference more specific to "(Kupper et al., 2015; see Fig. 4b therein)" and also clarify that the values given there are for total "grazing livestock" in Switzerland.

*2/25 was the model 'WindTrax' by ThunderBeach Scientific - need to cite model*
We did not use the model 'WindTrax' in the present study, but we used the model 'bLSmodelR' as described in Sect. 2.2.4.

*3/10 what was the topography (slope, barriers to flow, etc)*
The field site is generally flat with only a small slope towards South-West. There are no trees or hedges in the main wind sectors. The farm facilities north and south of the experimental field (Fig. 1) are the only barriers to the flow.

*4/12 were pressure and temperature corrections needed, if so give calibration factors*
No temperature or pressure corrections were needed within the given uncertainty range.

*4/12 was light intensity used to filter data, if so, give range*
As mentioned in Section 2.2.3 the miniDOAS measurements were filtered based on the level of light reaching the spectrometer. This led to a data rejection rate between about 1 % and 4 % for the different instruments.

*5/3 describe the model, and what modifications were made to Flesch's model, what was different*
We will add a reference to Häni et al. (2018), which has been published in the meantime (during the discussion phase). The model characteristics and the minor modifications to Flesch's original model are described there. The applied model 'bLSmodelR' itself was already used in other publications for $NH_3$ emission on pastures (Bell et al., 2017). But it has to be noted that we used the model without the newly introduced deposition module.

*5/15 however, the 'underlying' assumption of homogeneity of the emitting surface is less true with increased distance between the source and detector, please include this - it is unclear why the bLS model was not run in its entirety*
We are not sure whether we fully understand this referee comment. We measured close to the emitting surface (pasture paddock) and the pasture field has a generally small variability concerning the surface roughness (as reported by Felber et al., 2015, for the same site). The bLS model was run with a model domain of 250 m length, hence much larger compared to the actual emitting paddock. This will be clarified in the revised manuscript.

*5/25 state the given NH3 concentration certification*
The $NH_3$ percentage in the gas mixture had a relative uncertainty of 2%, i.e. the $NH_3$ mixing ratio was 5% ± 0.1%. We will add this information in the manuscript.

*5/31 'this is not necessarily the case' - this deserves further comment*

We will change the sentence to: "On a pasture cows can move freely and therefore the emission sources like urine and dung patches are usually not homogenous distributed and can lead to error prone emission estimates (Auerswald et al., 2010; Bell et al., 2017; Laubach et al., 2013a)."

*6/1 top page 11 states urine patches are the most important factor - then two ways of trying to estimate where these patches exit is tried by GPS of the dung piles and by locating the position of the cows - this cannot be direct emission map of ammonia since cows do not necessarily defecate and urinate at the same location, and the position of the cow adds little information to estimate urine patches.*

The spatial density distribution of urine and dung patches are not identical but very similar on a pasture (Auerswald et al., 2010). The miniDOAS line sensors integrate over a sufficient number of dung and urine patches, but measurement footprint only covers a part of the oblong paddocks. On some stages of the grazing season we could identify clear density gradients along the main paddock axis (see Fig. 9) with a generally high linear correlation between the distributions of dung and cow positions on the pasture. ($R^2$=0.98, see P11 L2). The fitted linear regression was used to estimate missing dung distributions and hence estimate the urine patch distribution for certain rotations. We will add a more detailed description of the procedure in the method section (see response to Referee#2, Comment 4 for details).

*6/7 it is not clear that the error would be reduced by compounding the errors in locating the urine patches, as opposed to assuming a uniform distribution, especially when the uniform criteria declines in importance with some distance downwind.*

As mentioned in the previous comment, we are quite sure that the information on the dung distribution can be used to estimate the distribution of the urine patches. As explained in the response to Comment 1 (see above), we could not have placed our sensors further downwind as we would have lost the possibility to observe the temporal behavior of the emissions as well as the sensitivity of the method (increase in concentration downwind of the paddock).

*7/27 need to expand by providing information on what was done regarding the bLS footprint*

This sentence was misleading because the bLS footprint was not directly used in the flux calculation. We will rephrase the sentence to:

"The field scale fluxes were determined based on the concentration differences of the paired MD systems and the dispersion coefficient *D* (see Eq. 1) computed by the bLS model."

*7/31 is this 50-70 hours per week?*

As shown in the referenced Table 1 the 50–70 hours correspond to the grazing duration on the investigated paddocks X.11 and X.12 per individual rotation.

*1/8 what is a 'strong' data filter - need to rewrite*

We will rephrase and refer to the data filtering criteria described in Sect 2.2.3.

*8/8 explain where the value '2.54' came from*

We will rephrase this paragraph to make it more clear to the reader (see also response to Comment 1 of Referee #2). Because of the low amount of available nighttime data, it was not possible to derive default emission curves for longer nighttime gaps (as shown for daytime conditions in Fig. 5). Thus it was assumed that the general temporal pattern is similar to daytime conditions but with a lower

amplitude for nighttime. The corresponding reduction factor (= 0.39) was based on the overall ratio between mean daytime and nighttime emissions during grazing.

*8/17 are you saying that your design, at specific wind directions, caused an interference of the incoming concentration (upwind) measurement which lead to an under-estimate of emissions - why not filter out the estimates?*
We cannot filter out those periods, as the investigated paddocks were part of an intensive rotational grazing system. This means upwind grazing took place frequently after grazing on the investigated paddocks between the miniDOAS systems. Filtering out those periods would lead to an unacceptable data loss. Additionally the interference effect is relatively small as shown in Fig. 6 (grey line) and Fig. 7 (red boxes). We also presented a way to correct for this effect (P8 L23 - 28). The interference effect has to be considered as a small disadvantage of an experimental design, which was optimized to fulfill several other requirements (see discussion in Section 3.6).

*9/3 use 'recorded' not 'retrieved'*
As the cumulative emissions are also based on gap filled data, we think 'recorded' is not suitable here. Therefore we would like to keep it unchanged.

*9/4 use 'greatest air temperature' and 9/5 'greater emissions'*
After consulting a native English speaker, we would like to keep 'highest' instead of 'greatest'.

*9/6 neither grazing duration nor N input is found in Table 3 - where are these data?*
Table 3 provides information on N input (separated into N excretion total and N excretion urine). Grazing duration can be found in Table 1. We will refer to Table 1 for grazing duration in the revised manuscript.

*10/10 usually as an alternative to mass flow controller, the entire tank is weight before and after, was this done in this study?*
We did not weight the tracer gas cylinder before and after the releases. But we used a sophisticated mass flow controller and checked its performance by measuring the individual orifices as described at P10 L7–11.

*10/16 do you mean 'air pressure'*
No, we mean the pressure within the tube of the artificial source system (between the gas tank and the flow controller). We will rephrase to '… the dynamic pressure within the tubes of the system upstream of the flow controller at the beginning …'.

*10/16 don't understand the set-up, what was no longer air tight - needs clarification, also need to indicate why air pressure is involved in recovery*
Similar to the previous answer, we did not mean air pressure but the pressure within the tracer gas tubing system. However, the proposed possible explanation for the high recovery rate in the first gas release trial was purely hypothetical. For clarity reasons we will remove it from the manuscript and state that we have no conclusive explanation for this individual result.

*10/19 'an unknown major error source is unlikely' - what does this mean, if unknown how can it be unlikely, delete this sentence as it adds no information - were the results used to correct the emission*

*or was it used to characterize the data? How sure are you that the difference was systematic, if this is important there needs to be a t-test done and if different then an accuracy analysis preformed to break the difference into systematic, random and slope errors*

We agree with the referee that the mentioned sentence is not useful and therefore we will omit it. With the artificial source we intended to test the applied methodology against a controlled source in an exemplary way, and it was not intended for a calibration or quantitative correction of the measurements. This will be clarified in the revised manuscript. As the artificial source experiments resulted in an average recovery rate that was not significantly different from 100 % (111 % ± 18 %) we assume that the used methodology (bLS dispersion modelling, concentration measurements with miniDOAS line sensors) was suitable for quantification of the pasture emissions.

If there exist minor systematic errors in the methodology (within the achieved uncertainty range, see Section 3.3.1), they are supposed to be very similar for both parallel pasture systems, and therefore do hardly affect the detection of differences between the two pasture systems (see  P11 L21-22, P12 L29-31).

*11/11 how was this correction done in all systems except system G rotation2, needs clarification - also need to document what this means for this latter value that was not corrected*

We are aware that the presentation of this correction procedure was not clear enough. We will therefore modify and enhance the corresponding method section. More details are given in the response to Referee#2 (Comment 4). We will also add the individual uncertainty ranges in Fig. 10.

*11/11 use 'greater uncertainty'*

After consulting a native English speaker, we prefer to leave the expression unchanged.

*12/3 cited reference not listed*

The cited reference to Móring et al. (2016) is listed correctly (P16 L8).

*13/6 delete 'under real practice conditions'*

*We would like to keep the sentence unchanged as previous studies on ammonia emissions* (e.g. Laubach et al., 2012, 2013) *were often not performed under realistic pasture conditions or included manual (artificial) application of urine to the soil.*

Auerswald, K., Mayer, F. and Schnyder, H.: Coupling of spatial and temporal pattern of cattle excreta patches on a low intensity pasture, Nutr. Cycl. Agroecosystems, 88(2), 275–288, doi:10.1007/s10705-009-9321-4, 2010.

Bell, M., Flechard, C., Fauvel, Y., Häni, C., Sintermann, J., Jocher, M., Menzi, H., Hensen, A. and Neftel, A.: Ammonia emissions from a grazed field estimated by miniDOAS measurements and inverse dispersion modelling, Atmospheric Meas. Tech., 10(5), 1875–1892, doi:10.5194/amt-10-1875-2017, 2017.

Häni, C., Flechard, C., Neftel, A., Sintermann, J. and Kupper, T.: Accounting for Field-Scale Dry Deposition in Backward Lagrangian Stochastic Dispersion Modelling of $NH_3$ Emissions, , doi:10.20944/preprints201803.0026.v1, 2018.

Kupper, T., Bonjour, C. and Menzi, H.: Evolution of farm and manure management and their influence on ammonia emissions from agriculture in Switzerland between 1990 and 2010, Atmos. Environ., 103, 215–221, doi:10.1016/j.atmosenv.2014.12.024, 2015.

Laubach, J., Taghizadeh-Toosi, A., Sherlock, R. R. and Kelliher, F. M.: Measuring and modelling ammonia emissions from a regular pattern of cattle urine patches, Agric. For. Meteorol., 156, 1–17, doi:10.1016/j.agrformet.2011.12.007, 2012.

Laubach, J., Bai, M., Pinares-Patiño, C. S., Phillips, F. A., Naylor, T. A., Molano, G., Rocha, E. A. C. and Griffith, D. W.: Accuracy of micrometeorological techniques for detecting a change in methane emissions from a herd of cattle, Agric. For. Meteorol., 176, 50–63, 2013a.

Laubach, J., Taghizadeh-Toosi, A., Gibbs, S. J., Sherlock, R. R., Kelliher, F. M. and Grover, S. P. P.: Ammonia emissions from cattle urine and dung excreted on pasture, Biogeosciences, 10(1), 327–338, doi:10.5194/bg-10-327-2013, 2013b.

---

## Author Comment (AC2) · 25 Jun 2018

**Author response to comments of referee #2**

We'd like to thank reviewer #2 for his answer and appreciate his valuable comments.

(referee comments are printed in *italic*, author responses are printed in blue)

*1. The methodology used for gap-filling emissions during night-time (and to a lesser extent for wind sectors coming from the surrounding farms) may be questionable. Indeed, the authors assume that night-time fluxes may be gap-filled based on day-time fluxes, but ammonia fluxes are fundamentally based on thermodynamical equilibrium at the surface (gas-liquid and acid-base equilibriums). This means that (1) the surface ammonia concentration is exponentially increasing with surface temperature due to the gas-liquid or Henry equilibrium, and we would hence expect lower emission at night due to lower temperature and hence lower concentration at the surface; (2) similarly, since ammonia fluxes are proportional to a concentration difference between the surface and a reference level, lower turbulent exchanges at night are expected to decrease night-time ammonia emissions. This means that using daytime fluxes amplitude and dynamics may systematically bias the emissions towards higher values. I would recommend a discussion on that point which may include a study on the temperature and u\* dependency of the ammonia emissions. Authors could refer to e.g. Flechard et al. (2013) for details on ammonia the points raised above.*

We fully agree with the reviewer that the ammonia emission depends on temperature and turbulence intensity and therefore is generally lower during nighttime compared to daytime conditions. We show this effect for the present study in Fig. 4a. We also account for the day-night difference in the gap filling procedure, but there was obviously a misinterpretation by the referee in this respect. Actually, we did not use the daytime fluxes amplitude to gap fill the missing night time fluxes. We only used the shape of the daytime curve (linear increase during grazing and exponential decrease afterwards), but with a significantly reduced amplitude by a factor of 1/2.54 during nighttime (see P8 L8). This factor was calculated from the ratio of available daytime and night time fluxes during the grazing phase. We will rephrase the first part of Section 3.2 to clarify the applied gap filling procedure and better explain the difference between daytime and nighttime cases.

Regarding the discussion about $u_*$ and temperature dependency, we already included this effects in our discussion and figures (P7 L28-30, P8 L9-10, Fig. 4) in a qualitative way. Due to the similar temporal pattern of $u_*$ (wind speed) and temperature at the study site and the frequent calm nighttime conditions it is unfortunately not possible (with a high degree of confidence) to disentangle the dependencies further. Yet we will include a reference to Flechard et al. (2013) stating why $NH_3$ emissions tend to be lower during nighttime conditions.

*2. Reference to the work of Moring et al. (2016) is lacking. In reference to this work, I wonder if considering the source as a mosaic of emission and deposition hot-spots rather than a distribution of emission patches would conceptually change the results presented here. Could the author elaborate on this question?*

There is actually a reference to Móring et al. (2016) in the manuscript (P12 L3). But we assume that the referee wanted to point towards the issue of simultaneous emission (from the excreta patches) and deposition (on the remaining pasture area) on the pasture field. In this respect, our measured fluxes represent the effective net $NH_3$ flux attributable to the grazing excreta (combination of emission

and re-deposition within the measured paddocks). But it does not include the large-scale background deposition, because the latter would not produce a horizontal concentration difference. Due to conceptual and practical reasons, a partitioning into gross emission and re-deposition was not in the scope of the present study. This would require separate measurements (e.g. by small-scale enclosures) of individual patches and of surrounding depositing surface areas. We will mention these issues in the revised manuscript (Section 2.2.4). In this context, we will reference Móring et al. (2017), where a simplified combination of modelled pasture emission and deposition is presented (while Móring et al., 2016 only presented a model for urine patch emission).

Concerning the artificial source experiment, the effect of re-deposition is presumably small as the downwind concentration was measured at only 6 m distance from the release line. Nevertheless, there might be a small bias towards lower recovery rates. As Häni et al. (2018) showed with a similar artificial release, but with $NH_3$ measurements at 15 m distance from the source, the dry deposition near the patches may be in the range of 10 %. We therefore assume that the error would be smaller in our experiment.

*3. The uncertainty analysis requires more details and especially on the gap-filling of emissions using the standard curve on Figure 5. An example showing a reconstructed emission would be beneficial here. It is difficult also to understand if the uncertainty analysis on gap-filling spans the actual variability in the fluxes shown in Figure 5 (the error-bars in Figure 5 would also need to be explained).*

An example of a gap filled time series (black points = reconstructed half-hourly data) is actually shown in Fig. 6. Our relatively simple gap filling approach is mainly based on interpolation between available data (either direct linear interpolation or with the help of the management related curves in Fig. 5). Therefore, a simple comparison between gap filled and measured data is not possible.

The uncertainty analysis in Fig. 7 is treating the (systematic) uncertainty of the cumulative emissions of an individual rotation, and not of half-hourly fluxes. Only the first error source ($\Delta C\_bias$) directly results from the systematic errors of the individual measurements. The other relevant error sources in Fig. 7 result from gap filling of missing flux values.

The vertical bars in Fig. 5 indicate the standard deviation of the half-hourly measurements within the 6-hour averaging interval. We will add this information in the figure caption. This variability does not represent an uncertainty but rather the variability in time (mainly between different rotations).

*4. The methodology used to derive the dung patches distribution, the way relative deviation of this dung patches Is calculated and the way it is used to correct cow based emissions all need clarifications. I think authors should mention previous work on patches emissions by Moring et al. (2016).*

The calibration/validation of the patch emission model in Móring et al. (2016) was based on an experiment with a defined pattern of artificially applied urine patches by Laubach et al. (2012). Therefore, we do not see how we can relate our assessment of real grazing patch distribution to their work. For other references to work of Móring et al. (2016) see response to Comment 2.

However, we agree with the referee that our methodology for determining patch distributions and correcting for their effect should be presented in a better way. In order to achieve this, we will modify and enhance Section 2.3 (Cow and excreta distribution monitoring) in the following way:

"The measured concentration difference and thus the derived $NH_3$ flux is mainly related to the emission of the surface area between the MD sensor paths on each grazing system (according to the main wind directions, see Fig. 1). This is only a part of the entire paddock area, which was considered as uniformly emitting area in the bLS calculations (Sect. 2.2.4) and for which the average urine N

input was quantified (Sect. 2.4). On pasture paddocks the cows can move freely and therefore the urine and dung patches may not be homogenously distributed on the entire area, which can lead to error prone emission estimates (Auerswald et al., 2010; Bell et al., 2017; Laubach et al., 2013). In order to assess the spatial distribution of the cow excreta on the paddocks X.11 and X.12 as main emission sources in our experiment, we used two different approaches. The number and position of dung patches was determined with a hand held GPS device within the first 3–5 days after grazing. In addition, the cow positions on the pasture were monitored with a day–night digital camera system at a temporal resolution of 10 minutes. The location of the individual cows were manually marked on the displayed pictures in a post-processing step. However, the night mode often did not yield useful information and therefore images showing the cow positions during nighttime were very sparse.

In order to account for inhomogeneity of the excreta distribution within the investigated paddocks, they were divided as shown in Fig. 3(new). The middle sections between the paired MD sensor paths represent the main source areas of the measured fluxes. Their excreta density $d_{X.meas}$ was related to the density of the entire paddocks $d_{(X.11+X.12)}$ to determine the excreta density correction factor $k_d$:

$$k_d = \frac{d_{(X.11+X.12)}}{d_{X.meas}}$$
(Eq. 2)

The exemplary dung patch survey in Fig. 3a(new) shows a positive deviation from the average paddock-wide density for both system M ($k_d$ = 1.28) and system G ($k_d$ = 1.40). However, dung observations were only available for two rotations for the paddock M.11, three rotations for G.11 and two rotations for X.12 while daytime cow position observation by camera was available for the whole measurement campaign for system M, and from rotation three onwards for system G. Missing dung density data were estimated from cow density distributions based on a regression analysis ($R^2$ = 0.98) between parallel surveys of density anomalies for dung patches and cow positions (Fig. 3b,new). Dung patch and cow position showed a very similar relative distributions with only a small offset. The excreta density anomaly factors $k_d$ (Eq. 2) derived from the combined information of the dung patch and the cow position surveys, were thus used to relate the observed cumulative NH$_3$ emissions to the entire paddock area and to the cow herd.

[Figure]

Figure 3 new: (a) GPS tagged dung positions recorded after grazing rotation 7 overlaid on a Google Earth image of the experimental area (Map data: Google, DigitalGlobe). The positions of the MD ammonia sensors/paths are indicated by the red dots/dotted lines. The white lines enclose the main emission measurement area between the sensors. Their dung patch

density $d_{X.meas}$ was related to the average density over the investigated paddocks according to Eq. 2; (b) comparison of $k_d$ values according to Eq. 2 for dung patch and cow position distributions on system M (blue) and system G (green)

*5. The description of the artificial source quality would benefit from more details on the homogeneity of the emissions, the pressure and flow rates stability. Some more examples on measured concentration and retrieved emissions during these trials would also be beneficial as these data were not published previously (to my knowledge).*
In the revised manuscript we will provide more information on the concentrations (mean+std) and absolute emissions of the individual experiments in Table 4. In addition we will provide information on the pressure and flow rate (and their stability) during the release.

*Detailed comments:*
*P4 L7: delete "important" before reference spectrum*
Will be changed accordingly.

*P4 L14-20: some details on the meteorological instruments may be useful. Please evaluate also how important may be high frequency losses on u* and H with 10Hz acquisition at 2 m above ground.*
Weather parameters like wind speed, precipitation, temperature and barometric pressure were measured with a WXT520 (Vaisala, Vantaa, FL). Global radiation was measured with a pyranometer (CNR1, Kipp&Zonen, Delft, NL). High frequency losses on $u_*$ and H due to the 10 Hz acquisition at 2 m above ground were typically below 5 %.

*P4 L25-30: 44%-49% missing values for low ustar may actually bias the analysis (see major comments)*
See response to Comment 1. The numbers indicated by the referee are total data loss due to wind direction and $u_*$ filtering. We will better specify the individual effects of $u_*$ filtering (26% and 30% for system M and G) in the revised manuscript. Additionally a potential 50% bias in (mainly nighttime) gap filled values is included in the uncertainty analyses.

*P5 L6: please give number of thousands of trajectory*
We used 25000 trajectories per line point (2 m apart).

*P5 L9: please give units of E, C and D. I woud also suggest to explicit the hypothesis behind this equation: actually Cdown = D*S + Cup, where not other nearby sources are assumed.*
We will add the units in the text, but we prefer not to change the equation because it directly represents the emission determination in the present study. Additionally the used form is consistent to other publications (Bell et al., 2017; Flesch et al., 2004)

*P5 L15-18: are the two hypothesis of a uniform and continuous distribution and of a random uniform distribution of sources strictly identical for inverse dispersion?*
Since our method is based on line integrating (i.e. line averaging) concentration measurements, we assume that the two hypothesis are equivalent as long as the footprint of the line concentration is large enough to cover a relatively large number of patches (as it was the case here).

*P5 L31: the work from Moring et al. (2016) should be referred to here and after.*
See response to Comment 4 above.

*P6 L4-8: the method used correct the cow-based emissions based on images and GPS needs to be detailed, as suggested in the major comment section.*
We agree. See response to Comment 4 above.

*P8 EQ-2: The meaning of this is unclear. $\Delta C_{UD}$ and Dup and Ddown should also be time dependent. Please clarify. I would suggest rather using t and Edef, i(t) etc.*
We will modify the text and the axis labelling in Fig. 5 to clarify that $t$ used in this equation is not the absolute measurement time but the elapsed time since the end of grazing of the individual upwind paddocks $i$.

*P9 L11-15: I would suggest showing the concentration inter-comparison figure in a supplementary material.*
The concentrations during the inter-comparison were typically very low at the remote station as the main focus was on retrieving the bias between the instruments. Therefore, we think that a corresponding figure would not provide useful additional information.

*P9 L28: please explain what is a "systemic" uncertainty.*
This is a typo and should be "systematic" uncertainty.

*P10 L2: I am not sure the word "stable" is appropriate here as it may be understood as "stable thermal stratification".*
We agree and will change the word to "stationary".

*P10 L7-10: I wonder what is the variability in the release rate between the critical orifice. I also wonder if the atmospheric pressure has an influence on the release rate at 30 minutes but also over short time scales (seconds). Finally, what is the expected (or even recorded) effect of wind speed variations on release rates : would one expect some ventury effects on the release rates?*
See also response to Comment 5 above. We think the Ventury effect is rather small as wind speeds at the height of the orifices (few cm above ground) were usually low. Additionally it would have no influence on the gas release as the mass flow controller would compensate for pressure fluctuations.

*P10 L28-32: It is quite unclear what the "relative deviation of the dung density" really is. I would suggest providing the exact equation.*
See response to Comment 4. We will add the exact equation.

*P11 L1-3: it is unclear how exactly missing values are obtained from regression analysis. Could the authors elaborate on that?*
See response to Comment 4 above. The regression analysis is illustrated in the new Fig. 3b displayed there.

*P11 L6-12: I would suggest giving details on how the uncertainties are aggregated (may be an equation in a supplementary section?)*
See response to Comment 3 above.

*Table 2: I would suggest finding a way to separate more clearly G and M in this table as in Table 3*
We will try to find a better way to separate the systems.

*Table 3: I suggest only proving 1 digit for temperature and none for rainfall.*
We agree with the reviewer and will change the entries accordingly.

*Figure 5: could you specify the meaning of the error-bars. I would also suggest using negative time values on the left.*
The error bars indicate the standard deviation of the measurements within the 6-hour period. We will add this information in the figure caption. We will modify the x-axis according to the referee suggestion.

*Figure 8: Could you provide error bars on both released and inverse modelling with measurements. I would also suggest changing one*
We will add the uncertainties of the measurements as error bars.

*Figure 10: Please explicit the term "relative deviation" in the legend.*
We will insert a reference to the equation as described in the response to Comment 4 above.

*Figure 11. Please explicit if error bars are standard deviation, standard errors or interquartile*
We will include an explanation of the error bars in the figure caption.

*References:*
Auerswald, K., Mayer, F. and Schnyder, H.: Coupling of spatial and temporal pattern of cattle excreta patches on a low intensity pasture, Nutr. Cycl. Agroecosystems, 88(2), 275–288, doi:10.1007/s10705-009-9321-4, 2010.

Bell, M., Flechard, C., Fauvel, Y., Häni, C., Sintermann, J., Jocher, M., Menzi, H., Hensen, A. and Neftel, A.: Ammonia emissions from a grazed field estimated by miniDOAS measurements and inverse dispersion modelling, Atmospheric Meas. Tech., 10(5), 1875–1892, doi:10.5194/amt-10-1875-2017, 2017.

Flesch, T. K., Wilson, J. D., Harper, L. A., Crenna, B. P. and Sharpe, R. R.: Deducing Ground-to-Air Emissions from Observed Trace Gas Concentrations: A Field Trial, J. Appl. Meteorol., 43(3), 487–502, doi:10.1175/1520-0450(2004)043<0487:DGEFOT>2.0.CO;2, 2004.

Häni, C., Flechard, C., Neftel, A., Sintermann, J. and Kupper, T.: Accounting for Field-Scale Dry Deposition in Backward Lagrangian Stochastic Dispersion Modelling of $NH_3$ Emissions, doi:10.20944/preprints201803.0026.v1, 2018.

Laubach, J., Taghizadeh-Toosi, A., Sherlock, R. R. and Kelliher, F. M.: Measuring and modelling ammonia emissions from a regular pattern of cattle urine patches, Agric. For. Meteorol., 156, 1–17, doi:10.1016/j.agrformet.2011.12.007, 2012.

Laubach, J., Bai, M., Pinares-Patiño, C. S., Phillips, F. A., Naylor, T. A., Molano, G., Rocha, E. A. C. and Griffith, D. W.: Accuracy of micrometeorological techniques for detecting a change in methane emissions from a herd of cattle, Agric. For. Meteorol., 176, 50–63, 2013.

Móring, A., Vieno, M., Doherty, R. M., Laubach, J., Taghizadeh-Toosi, A. and Sutton, M. A.: A process-based model for ammonia emission from urine patches, GAG (Generation of Ammonia from Grazing): description and sensitivity analysis, Biogeosciences, 13(6), 1837–1861, doi:10.5194/bg-13-1837-2016, 2016.

Móring, A., Vieno, M., Doherty, R. M., Milford, C., Nemitz, E., Twigg, M. M., Horváth, L. and Sutton, M. A.: Process-based modelling of NH3 exchange with grazed grasslands, Biogeosciences, 14(18), 4161–4193, doi:10.5194/bg-14-4161-2017, 2017.

---

## Author Response (AR1)

bg-2018-86

**Author response to comments of referee #1**

We'd like to thank reviewer #1 for his answer and appreciate his valuable comments.

(referee comments are printed in *italic*, author responses are printed in blue)

*1. The weakest part of the study was investigation in estimating urine patch locations to avoid assumptions on the uniformity of the source on pasture (by measuring dung pile locations and position of the cows). In applications where bLS is often used in non-uniform sources, it is realized that the detector should be some distance downwind to minimize the impact of non-uniform source on emissions but close enough to resolve horizontal gradient (elevated background concentration, a possible problem in this study).*

We do not really understand this statement. In our view, the estimation of the excreta distribution on a real grazed pasture is, despite the necessary approximations,  one of the strength of the present study since this issue is either missing in comparable studies (Bell et al., 2017), artificially forced by distributing urine manually on the pasture (Laubach et al., 2012) or by forcing unrealistically high excreta densities during short experiments (Laubach et al., 2013b). We made dung patch surveys and we applied a robust method to estimate dung patch densities based on visual cow monitoring with camera systems. As pointed out in the manuscript, it is a valid assumption that urine and dung patches are similar distributed on the paddock (P12 L28). Auerswald et al. (2010) also found a similar spatial distribution between urine and dung patches on a low intensity pasture.

We are fully aware that placing a detector further downwind minimizes the impact of a non-uniform source. Nevertheless, as referee #2 also pointed out, the small fields ensured a temporarily high stocking density and hence a good sensitivity of the concentration measurement method. We would loss this advantage if placing the detector further downwind. Additionally, for maintaining two realistic grazing systems over an entire season, it was not possible to keep the animals in smaller paddocks. As we investigated a rotational management, placing the detector further downwind would have resulted in an average emission measurement over multiple paddocks (or differently managed areas). Therefore monitoring the temporal dynamics of the emissions (increase with grazing duration, exp. decrease afterwards) would not have been possible.

However, we realized that the presentation of this issue in the manuscript was not optimal and this may have influenced the referee comment. Therefore we improved the manuscript in this respect (see specific comments below).

*Detailed comments:*
*For the majority of the minor (mostly language related) comments we followed the referee suggestions. Here only the comments that need an answer are listed.*

*1/15 insert 'maximum of x µg N-NH3 m-2 s-1 at the end'*
As the maximum emissions at the end of the grazing period varied (mostly due to different grazing duration), we kept the sentence unchanged. The overall maximum flux value is included in the range given in the previous sentence.

*2/5 'about eight times lower' - could not find this in Kupper et al (2015) - re-check citation*
The 'eight times lower' factor was calculated based on the TAN flows in Fig. 4b in Kupper et al. (2015). That figure shows that the relative $NH_3$ emission of grazing livestock (8.9% of excreta TAN) is 7.6 times lower compared to indoor housing including storage and spreading of manure (67.8% of excreta TAN). This factor was rounded to 'about eight'.
We made the reference more specific to "… model Agrammon (Kupper et al., 2015; see Fig. 4b therein) grazing livestock produces …".

*2/25 was the model 'WindTrax' by ThunderBeach Scientific - need to cite model*
We did not use the model 'WindTrax' in the present study, but we used the model 'bLSmodelR' as described in Sect. 2.2.4.

*3/10 what was the topography (slope, barriers to flow, etc)*
The field site is generally flat with only a small slope towards South-West. There are no trees or hedges in the main wind sectors. The farm facilities north and south of the experimental field (Fig. 1) are the only barriers to the flow.

*4/12 were pressure and temperature corrections needed, if so give calibration factors*
No temperature or pressure corrections were needed within the given uncertainty range.

*4/12 was light intensity used to filter data, if so, give range*
As mentioned in Section 2.2.3 the miniDOAS measurements were filtered based on the level of light reaching the spectrometer. This led to a data rejection rate between about 1 % and 4 % for the different instruments.

*5/3 describe the model, and what modifications were made to Flesch's model, what was different*
We added a reference to Häni et al. (2018) in Sect. 2.2.4, which has been published in the meantime (during the discussion phase). The model characteristics and the minor modifications to Flesch's original model are described there. Additionally we provided more details on the model. The applied model 'bLSmodelR' itself was already used in other publications for $NH_3$ emission on pastures (Bell et al., 2017). But it has to be noted that we used the model without the newly introduced deposition module.

*5/15 however, the 'underlying' assumption of homogeneity of the emitting surface is less true with increased distance between the source and detector, please include this - it is unclear why the bLS model was not run in its entirety*
We are not sure whether we fully understand this referee comment. We measured close to the emitting surface (pasture paddock) and the pasture field has a generally small variability concerning the surface roughness (as reported by Felber et al., 2015, for the same site). The bLS model was run with a model domain of 250 m length, hence much larger compared to the actual emitting paddock. This has been clarified in the revised manuscript.

*5/25 state the given NH3 concentration certification*
The $NH_3$ percentage in the gas mixture had a relative uncertainty of 2%, i.e. the $NH_3$ mixing ratio was 5% ± 0.1%. We added this information in the manuscript.

*5/31 'this is not necessarily the case' - this deserves further comment*

We chanced the sentence to: "On a pasture cows can move freely and therefore the urine and dung patches may not be homogenously distributed on the entire area, which can lead to error prone emission estimates (Auerswald et al., 2010; Bell et al., 2017; Laubach et al., 2013a)."

*6/1 top page 11 states urine patches are the most important factor - then two ways of trying to estimate where these patches exit is tried by GPS of the dung piles and by locating the position of the cows - this cannot be direct emission map of ammonia since cows do not necessarily defecate and urinate at the same location, and the position of the cow adds little information to estimate urine patches.*

The spatial density distribution of urine and dung patches are not identical but very similar on a pasture (Auerswald et al., 2010). The miniDOAS line sensors integrate over a sufficient number of dung and urine patches, but measurement footprint only covers a part of the oblong paddocks. On some stages of the grazing season we could identify clear density gradients along the main paddock axis (see Fig. 3 in the revised manuscript) with a generally high linear correlation between the distributions of dung and cow positions on the pasture. ($R^2$=0.98, see P11 L2). The fitted linear regression was used to estimate missing dung distributions and hence estimate the urine patch distribution for certain rotations.

We added a more detailed description (incl. equations) of the procedure in the method section 2.4, and added the new Fig. 3 to illustrate the applied method. Fig. 3a (formerly Fig. 9) was modified, and Fig. 3b shows the regression between parallel surveys of density anomalies for dung patches and cow positions.

*6/7 it is not clear that the error would be reduced by compounding the errors in locating the urine patches, as opposed to assuming a uniform distribution, especially when the uniform criteria declines in importance with some distance downwind.*

As mentioned in the previous comment, we are quite sure that the information on the dung distribution can be used to estimate the distribution of the urine patches. As explained in the response to Comment 1 (see above), we could not have placed our sensors further downwind as we would have lost the possibility to observe the temporal behavior of the emissions as well as the sensitivity of the method (increase in concentration downwind of the paddock).

*7/27 need to expand by providing information on what was done regarding the bLS footprint*

This sentence was misleading because the bLS footprint was not directly used in the flux calculation. We rephrased the sentence to:

"The field scale fluxes were determined based on the concentration differences of the paired MD systems and the dispersion coefficient $D$ (see Eq. 1) computed by the bLS model."

*7/31 is this 50-70 hours per week?*

As shown in the referenced Table 1, the 50–70 hours correspond to the grazing duration on the investigated paddocks X.11 and X.12 per individual rotation.

*1/8 what is a 'strong' data filter - need to rewrite*

We rephrased the sentence and referred to the data filtering criteria described in Sect 2.2.3.

*8/8 explain where the value '2.54' came from*
We rephrased this paragraph to clarify the issue (see also response to Comment 1 of Referee #2). Because of the low amount of available nighttime data, it was not possible to derive default emission curves for longer nighttime gaps (as shown for daytime conditions in Fig. 6 in the revised manuscript). Thus it was assumed that the general temporal pattern is similar to daytime conditions but with a lower amplitude for nighttime. The corresponding reduction factor (= 0.39, corresponds to the inverse of the original factor 2.54) was based on the overall ratio between mean nighttime and daytime emissions during grazing.

*8/17 are you saying that your design, at specific wind directions, caused an interference of the incoming concentration (upwind) measurement which lead to an under-estimate of emissions - why not filter out the estimates?*
We cannot filter out those periods, as the investigated paddocks were part of an intensive rotational grazing system. This means upwind grazing took place frequently after grazing on the investigated paddocks between the miniDOAS systems. Filtering out those periods would lead to an unacceptable data loss. Additionally the interference effect is relatively small as shown in Fig. 7 (grey line) and Fig. 8 (red boxes). We also presented a way to correct for this effect (P9 L30 - 34). The interference effect has to be considered as a small disadvantage of our experimental design, which was optimized to fulfill several other requirements (see discussion in Section 3.6).

*9/3 use 'recorded' not 'retrieved'*
As the cumulative emissions are also based on gap filled data, we think 'recorded' is not suitable here. Therefore we kept it unchanged.

*9/4 use 'greatest air temperature' and 9/5 'greater emissions'*
After consulting a native English speaker, we kept 'highest' instead of 'greatest'.

*9/6 neither grazing duration nor N input is found in Table 3 - where are these data?*
Table 3 provides information on N input (separated into N excretion total and N excretion urine). Grazing duration can be found in Table 1. We referred to Table 1 for grazing duration in the revised manuscript.

*10/10 usually as an alternative to mass flow controller, the entire tank is weight before and after, was this done in this study?*
We did not weight the tracer gas cylinder before and after the releases. But we used a sophisticated mass flow controller and checked its performance by measuring the individual orifices as described at P11 L14–16.

*10/16 do you mean 'air pressure'*
No, we mean the pressure within the tube of the artificial source system (between the gas tank and the flow controller). We rephrased to '…During that particular release the dynamic pressure within the tubes of the system upstream of the flow controller …'.

*10/16 don't understand the set-up, what was no longer air tight - needs clarification, also need to indicate why air pressure is involved in recovery*

Similar to the previous answer, we did not mean air pressure but the pressure within the tracer gas tubing system. However, the proposed possible explanation for the high recovery rate in the first gas release trial was purely hypothetical. For clarity reasons we removed it from the manuscript and state that we have no conclusive explanation for this individual result.

*10/19 'an unknown major error source is unlikely' - what does this mean, if unknown how can it be unlikely, delete this sentence as it adds no information - were the results used to correct the emission or was it used to characterize the data? How sure are you that the difference was systematic, if this is important there needs to be a t-test done and if different then an accuracy analysis preformed to break the difference into systematic, random and slope errors*

We agree with the referee that the mentioned sentence is not useful and therefore deleted it. With the artificial source we intended to test the applied methodology against a controlled source in an exemplary way, and it was not intended for a calibration or quantitative correction of the measurements. This was clarified in the revised manuscript.

As the artificial source experiments resulted in an average recovery rate that was not significantly different from 100 % (111 % ± 18 %) we assume that the used methodology (bLS dispersion modelling, concentration measurements with miniDOAS line sensors) was suitable for quantification of the pasture emissions. If there exist minor systematic errors in the methodology (within the achieved uncertainty range, see Section 3.3.1), they are supposed to be very similar for both parallel pasture systems, and therefore do hardly affect the detection of differences between the two pasture systems (see  P12 L17-19, P13 L27-29).

*11/11 how was this correction done in all systems except system G rotation2, needs clarification - also need to document what this means for this latter value that was not corrected*

We are aware that the presentation of this correction procedure was not clear enough. We modified and enhanced the corresponding method section 2.4. More details are given in the response to Referee#2 (Comment 4). We also added the individual uncertainty ranges for the single rotations in the modified Fig. 10.

*11/11 use 'greater uncertainty'*

After consulting a native English speaker, we left the expression unchanged.

*12/3 cited reference not listed*

The cited reference to Móring et al. (2016) was listed correctly (P17 L5) in the discussion paper.

*13/6 delete 'under real practice conditions'*

*We kept the sentence unchanged as previous studies on ammonia emissions* (e.g. Laubach et al., 2012, 2013) *were often not performed under realistic pasture conditions or included manual (artificial) application of urine to the soil.*

**Author response to comments of referee #2**

We'd like to thank reviewer #2 for his answer and appreciate his valuable comments.

(referee comments are printed in *italic*, author responses are printed in blue)

*1. The methodology used for gap-filling emissions during night-time (and to a lesser extent for wind sectors coming from the surrounding farms) may be questionable. Indeed, the authors assume that night-time fluxes may be gap-filled based on day-time fluxes, but ammonia fluxes are fundamentally based on thermodynamical equilibrium at the surface (gas-liquid and acid-base equilibriums). This means that (1) the surface ammonia concentration is exponentially increasing with surface temperature due to the gas-liquid or Henry equilibrium, and we would hence expect lower emission at night due to lower temperature and hence lower concentration at the surface; (2) similarly, since ammonia fluxes are proportional to a concentration difference between the surface and a reference level, lower turbulent exchanges at night are expected to decrease night-time ammonia emissions. This means that using daytime fluxes amplitude and dynamics may systematically bias the emissions towards higher values. I would recommend a discussion on that point which may include a study on the temperature and u\* dependency of the ammonia emissions. Authors could refer to e.g. Flechard et al. (2013) for details on ammonia the points raised above.*

We fully agree with the reviewer that the ammonia emission depends on temperature and turbulence intensity and therefore is generally lower during nighttime compared to daytime conditions. We show this effect for the present study in Fig. 5a. We also account for the day-night difference in the gap filling procedure, but there was obviously a misinterpretation by the referee in this respect. Actually, we did not use the daytime fluxes amplitude to gap fill the missing night time fluxes. We only used the shape of the daytime curve (linear increase during grazing and exponential decrease afterwards), but with a significantly reduced amplitude (0.39) during nighttime (see P9 L12-14). This factor was calculated from the ratio of available nighttime and daytime fluxes during the grazing phase. We rephrased Sect. 3.2 to clarify the applied gap filling procedure and better explain the difference between daytime and nighttime cases.

Regarding the discussion about $u_*$ and temperature dependency, we already included this effects in our discussion and figures (P8 L26-28, P9 L19-21, Fig. 5) in a qualitative way. Due to the similar temporal pattern of $u_*$ (wind speed) and temperature at the study site and the frequent calm nighttime conditions it is unfortunately not possible (with a high degree of confidence) to disentangle the dependencies further.

We included a reference to Flechard et al. (2013) (P8 L 32) stating why $NH_3$ emissions tend to be lower during nighttime conditions.

*2. Reference to the work of Moring et al. (2016) is lacking. In reference to this work, I wonder if considering the source as a mosaic of emission and deposition hot-spots rather than a distribution of emission patches would conceptually change the results presented here. Could the author elaborate on this question?*

There was already a reference to Móring et al. (2016) in the manuscript (P12 L33). But we assume that the referee wanted to point towards the issue of simultaneous emission (from the excreta patches) and deposition (on the remaining pasture area) on the pasture field. In this respect, our measured fluxes represent the effective net $NH_3$ flux attributable to the grazing excreta (combination of emission and re-deposition within the measured paddocks). But it does not include the large-scale background deposition, because the latter would not produce a horizontal concentration difference. Due to conceptual and practical reasons, a partitioning into gross emission and re-deposition was not in the scope of the present study. This would require separate measurements (e.g. by small-scale enclosures) of individual patches and of surrounding depositing surface areas.

We added a paragraph about this issue in the revised manuscript (Section 2.2.4, P 5 L 18 - 23). In this context, we referenced Móring et al. (2017), where a simplified combination of modelled pasture emission and deposition is presented (while Móring et al., 2016 only presented a model for urine patch emission).

Concerning the artificial source experiment, the effect of re-deposition is presumably small as the downwind concentration was measured at only 6 m distance from the release line. Nevertheless, there might be a small bias towards lower recovery rates. As Häni et al. (2018) showed with a similar artificial release, but with $NH_3$ measurements at 15 m distance from the source, the dry deposition near the point sources may be in the range of 10 %. We therefore assume that the error would be smaller in our experiment.

*3. The uncertainty analysis requires more details and especially on the gap-filling of emissions using the standard curve on Figure 5. An example showing a reconstructed emission would be beneficial here. It is difficult also to understand if the uncertainty analysis on gap-filling spans the actual variability in the fluxes shown in Figure 5 (the error-bars in Figure 5 would also need to be explained).*

An example of a gap filled time series (black points = reconstructed half-hourly data) is actually shown in Fig. 7. Our relatively simple gap filling approach is mainly based on interpolation between available data (either direct linear interpolation or with the help of the management related curves in Fig. 6). Therefore, a simple comparison between gap filled and measured data is not possible.

The uncertainty analysis in Fig. 8 is treating the (systematic) uncertainty of the cumulative emission of an individual rotation, and not of half-hourly fluxes. Only the first error source ($\Delta C\_bias$) directly results from the systematic errors of the individual measurements. The other relevant error sources in Fig. 8 result from gap filling of missing flux values.

The vertical bars in Fig. 6 indicate the standard deviation of the half-hourly measurements within the 6-hour averaging interval. We added this information in the figure caption. This variability does not represent an uncertainty but rather the variability in time (mainly between different rotations).

*4. The methodology used to derive the dung patches distribution, the way relative deviation of this dung patches Is calculated and the way it is used to correct cow based emissions all need clarifications. I think authors should mention previous work on patches emissions by Moring et al. (2016).*

The calibration/validation of the patch emission model in Móring et al. (2016) was based on an experiment with a defined pattern of artificially applied urine patches by Laubach et al. (2012). Therefore, we do not see how we can relate our assessment of real grazing patch distribution to their work. For other references to work of Móring et al. (2016) see response to Comment 2.

However, we agree with the referee that our methodology for determining patch distributions and correcting for their effect should have been presented in a better way. In order to achieve this in the revised manuscript, we modified and enhanced Section 2.4 (Cow and excreta distribution monitoring) in the following way:

i) We added a paragraph at the beginning of Sect. 2.4 clarifying the issue of homogenous or uniform source distribution within the paddock area, deviation from that ideal assumption and the effect on the bLS calculation.

ii) We moved the old Fig. 9 from to this section (new Fig. 3a) and improved it to illustrate the problem of excreta patch density distribution as well as the applied correction approach.

iii) We added the new Fig. 3b showing the linear relationship between density anomalies for dung patches and cow positions. This supports the use of cow position information in case of lacking dung patch surveys.

iv) We added to new equations (Eq. 2 and 3) in the second part of Sect. 2.4 describing the density correction factor $k_d$ and the calculation of the cumulative integral $NH_3$ emission $E_{int}$ for each rotation.

*5. The description of the artificial source quality would benefit from more details on the homogeneity of the emissions, the pressure and flow rates stability. Some more examples on measured concentration and retrieved emissions during these trials would also be beneficial as these data were not published previously (to my knowledge).*

In the revised manuscript we have included more information on the concentrations (mean+std) and absolute emissions of the individual experiments in Table 4. In addition we have included data on the pressure and flow rate (and their stability) during the release.

*Detailed comments:*

*P4 L7: delete "important" before reference spectrum*

Was changed accordingly.

*P4 L14-20: some details on the meteorological instruments may be useful. Please evaluate also how important may be high frequency losses on u\* and H with 10Hz acquisition at 2 m above ground.*

Weather parameters like wind speed, precipitation, temperature and barometric pressure were measured with a WXT520 (Vaisala, Vantaa, FL). Global radiation was measured with a pyranometer (CNR1, Kipp&Zonen, Delft, NL). We added this information in Sect. 2.2.2.

High frequency losses on $u_*$ and H due to the 10 Hz acquisition at 2 m above ground were typically below 5 %.

*P4 L25-30: 44%-49% missing values for low ustar may actually bias the analysis (see major comments)*

See response to Comment 1. The numbers indicated by the referee are total data loss due to wind direction and $u_*$ filtering. We better specified the individual effects of $u_*$ filtering (26% and 30% for system M and G) in the revised manuscript (Sect. 2.2.3). Additionally a potential 50% bias in (mainly nighttime) gap filled values is already included in the uncertainty analyses.

*P5 L6: please give number of thousands of trajectory*

We used 25'000 trajectories per line point (2 m apart). We added this information in Sect. 2.2.4 (P5, L4).

*P5 L9: please give units of E, C and D. I woud also suggest to explicit the hypothesis behind this equation: actually Cdown = D\*S + Cup, where not other nearby sources are assumed.*

We added the units in the text, but we preferred not to change the equation because it directly represents the emission determination in the present study. Additionally the used form is consistent to other publications (Bell et al., 2017; Flesch et al., 2004)

*P5 L15-18: are the two hypothesis of a uniform and continuous distribution and of a random uniform distribution of sources strictly identical for inverse dispersion?*

Since our method is based on line integrating (i.e. line averaging) concentration measurements, we assume that the two hypothesis are equivalent as long as the footprint of the line concentration is large enough to cover a relatively large number of patches (as it was the case here).

*P5 L31: the work from Moring et al. (2016) should be referred to here and after.*

See response to Comment 4 above.

*P6 L4-8: the method used correct the cow-based emissions based on images and GPS needs to be detailed, as suggested in the major comment section.*

We agree. See response to Comment 4 above.

*P8 EQ-2: The meaning of this is unclear. $\Delta C_{UD}$ and Dup and Ddown should also be time dependent. Please clarify. I would suggest rather using t and Edef, i(t) etc.*

We modified the text and the axis labelling in Fig. 6 to clarify that $t$ used in this equation is not the absolute measurement time but the elapsed time since the end of grazing of the individual upwind paddock $i$.

*P9 L11-15: I would suggest showing the concentration inter-comparison figure in a supplementary material.*

The concentrations during the inter-comparison were typically very low at the remote station as the main focus was on retrieving the bias between the instruments. Therefore, we think that a corresponding figure would not provide useful additional information.

*P9 L28: please explain what is a "systemic" uncertainty.*

This is a typo and should be "systematic" uncertainty. We changed it accordingly.

*P10 L2: I am not sure the word "stable" is appropriate here as it may be understood as "stable thermal stratification".*

We agree and changed the word to "stationary".

*P10 L7-10: I wonder what is the variability in the release rate between the critical orifice. I also wonder if the atmospheric pressure has an influence on the release rate at 30 minutes but also over short time scales (seconds). Finally, what is the expected (or even recorded) effect of wind speed variations on release rates : would one expect some ventury effects on the release rates?*

See also response to Comment 5 above. We think the Ventury effect is rather small as wind speeds at the height of the orifices (few cm above ground) were usually low. Additionally it would have no influence on the gas release as the mass flow controller would compensate for pressure fluctuations.

*P10 L28-32: It is quite unclear what the "relative deviation of the dung density" really is. I would suggest providing the exact equation.*

See response to Comment 4. We added Eq. 2 in Sect. 2.4.

*P11 L1-3: it is unclear how exactly missing values are obtained from regression analysis. Could the authors elaborate on that?*

See response to Comment 4 above. The regression analysis is illustrated in the new Fig. 3b.

*P11 L6-12: I would suggest giving details on how the uncertainties are aggregated (may be an equation in a supplementary section?)*
See response to Comment 3 above.

*Table 2: I would suggest finding a way to separate more clearly G and M in this table as in Table 3*
We modified the layout of Table 2 and Table 3 for better readability.

*Table 3: I suggest only proving 1 digit for temperature and none for rainfall.*
We agree with the reviewer and changed the entries accordingly.

*Figure 5: could you specify the meaning of the error-bars. I would also suggest using negative time values on the left.*
The error bars indicate the standard deviation of the measurements within the 6-hour period. We added this information in the figure caption (Fig. 6 in the revised manuscript). We modified the x-axis according to the referee suggestion.

*Figure 8: Could you provide error bars on both released and inverse modelling with measurements. I would also suggest changing one*
We added the uncertainties of the measurements as error bars (Fig. 9 in the revised manuscript).

*Figure 10: Please explicit the term "relative deviation" in the legend.*
We modified the whole figure for better readability. Additionally we inserted a reference to the equation (Eq. 2 and 3) as described in the response to Comment 4 above.

*Figure 11. Please explicit if error bars are standard deviation, standard errors or interquartile*
We included an explanation of the error bars in the figure caption.

*References:*

[revised manuscript text omitted]